# Aircraft measurements of water vapor heavy isotope ratios in the marine boundary layer and lower troposphere during ORACLES

Dean Henze[1], David Noone[1,2], Darin Toohey[3]

[1] College of Earth, Ocean, and Atmospheric Science, Oregon State University, OR, USA

[2] Department of Physics, University of Auckland, Auckland, New Zealand

[3] Department of Atmospheric and Ocean Sciences University of Colorado, CO, USA

*Correspondence to:* Dean Henze (henzede@oregonstate.edu), David Noone (david.noone@auckland.ac.nz)

**Abstract**. This paper presents aircraft in-situ measurements of specific humidity and heavy water isotope ratios D/H and $^{18}O/^{16}O$ during the NASA ObseRvations of Aerosols above CLouds and their intEractionS (ORACLES) project. The aircraft measurement system is also presented. The dataset is unique in that 1) it contains both vapor and cloud condensed water isotope ratios, 2) it spans sufficient space and time to enable construction of spatially resolved climatology of isotope ratios in the lower troposphere, and 3) is it paired with a wealth of complementary measurements on atmospheric thermodynamic, chemical, aerosol and radiative properties. Aircraft sampling took place in the southeast Atlantic marine boundary layer and lower troposphere (equator to 22˚S) over the months of Sept. 2016, Aug. 2017, and Oct. 2018. Isotope measurements were made using cavity ring-down spectroscopic analyzers integrated into the Water Isotope System for Precipitation and Entrainment Research (WISPER). From an isotope perspective, the 300+ hours of 1 Hz in-situ data at levels in the atmosphere ranging from 70 m to 7 km represents a remarkably large and vertically resolved dataset. This paper provides a brief overview of the ORACLES mission and describes how water vapor heavy isotope ratios fit within the experimental design. Overviews of the sampling region and sampling strategy are presented, followed by the WISPER system setup and calibration details. The three data formats available to the users are each covered (latitude-altitude curtains, individual vertical profiles, and timeseries), with illustrative examples to highlight some features of the dataset and provide a plausibility check. The curtains and profiles demonstrate the dataset's potential to provide a comprehensive perspective on moisture transport and isotopic content in this region. Finally, measurement uncertainties are provided. Curtain and vertical profile data for all sampling periods can be accessed at https://doi.org/10.5281/zenodo.5748368 (see Henze et al., 2022). Time series data for the Sept. 2016, Aug. 2017, and Oct. 2018 sampling periods can be accessed at https://doi.org/10.5067/Suborbital/ORACLES/P3/2016_V3, https://doi.org/10.5067/Suborbital/ORACLES/P3/2017_V3, and https://doi.org/10.5067/Suborbital/ORACLES/P3/2018_V3, respectively (see references for ORACLES Science Team, 2020 – 2016 P3 data, 2017 P3 data, and 2018 P3 data).

## 1. Introduction

Uncertainties in general circulation model (GCM) representations of marine boundary layer (MBL) low cloud cover contribute substantially to the spread in model predictions of future climate (Bony & Dufresne, 2005). Further uncertainties in GCM output arise from an incomplete understanding of cloud-aerosol interactions. For example, the Fifth Assessment Report from the Intergovernmental Panel on Climate Change (IPCC) identified cloud-aerosol interactions as the largest contribution to uncertainty in total radiative forcing estimates (Boucher et al., 2013 IPCC report, Chapter 7). Given these limitations, there is a need to develop a refined understanding of several key processes that control MBL low cloud cover.

The formation of marine low clouds is linked in part to the energy and moisture budgets of the marine boundary layer (Wood 2012; Vial et al., 2017), and there is a need for tighter observational constraints on these budgets. Measurements of the oxygen and hydrogen stable isotopic composition of atmospheric water vapor present a way to obtain tighter constraints since they provide information on the relative importance of air mass mixing, precipitation,

and other moisture transport processes not easy to determine using conventional thermodynamic variables alone (e.g. Risi et al., 2008; Brown et al., 2013; Galewsky et al., 2016). Therefore, water vapor isotope ratios can be utilized to constrain uncertainties in low cloud thermodynamics. Potentially, they could be applied to moisture transport and cloud microphysics processes associated with aerosol indirect effects, such as the lifetime effect and precipitation suppression. To date, detailed vertical profiles in the lower troposphere are sparse (e.g. Ehhalt, 1974; Herman et al., 2014; Dyroff et al., 2015). This limits the degree to which isotope ratios can be fully leveraged to provide a comprehensive depiction of the atmospheric water budget in the lower troposphere.

The NASA ObseRvations of Aerosols above CLouds and their intEractionS (ORACLES) mission provides an extensive data set of aircraft in-situ aerosol, cloud microphysical, water vapor heavy isotope ratio, and meteorological measurements in the southeast Atlantic (Fig. 1) during the months of Sept. 2016, Aug. 2017, and Oct., 2018 (Redemann et al., 2020). The southeast Atlantic (SEA) is an ideal region for cloud-aerosol effects research because seasonal biomass burning aerosol (BBA) plumes from the African continent subside onto a semi-permanent stratocumulus cloud deck (Adebiyi and Zuidema, 2016; Garstang et al., 1996), where they may entrain into the marine boundary layer. The study accumulated over 300 hours of in-situ measurements (corresponding to ~140,000 linear km for an airspeed of 250 knots) at 1 Hz frequency in the MBL and overlying troposphere from 70 m up to 6 km. The dataset captures numerous MBL states and cloud layers with varying degrees of BBA loading, as well as cases where the MBL was in contact with both high and low BBA loaded layers in the overlying troposphere.

Isotope ratios were collected with the new Water Isotope System for Precipitation and Entrainment Research (WISPER). The objective of this paper is to describe WISPER and the extensive isotope ratio dataset. Section 2 describes the sampling region and strategy, section 3 introduces WISPER, and section 4 covers calibration methods with additional details in appendices. Section 5 covers the three datasets available (latitude-altitude curtains, individual vertical profiles, and timeseries data), with illustrative examples highlighting some key features of the data set. Measurement uncertainties are given at the end of section 5. Section 6 provides a few summary remarks.

## 2. ORACLES study region and sampling strategy

An extensive overview of the ORACLES project is presented in Redemann et. al. (2021). Aspects of the project that provide relevant context for the isotopic datasets are outlined below. In-situ sampling aboard the NASA P3-Orion aircraft spanned the SEA MBL and lower troposphere (LT) during the agricultural burning season in subtropical southern Africa over southern hemisphere spring. During this season, BBA loaded air in the African PBL is carried out over the SEA by lower troposphere easterly flow (Adebiyi and Zuidema, 2016; Garstang et al., 1996). This air is brought over the SEA cloud deck due to large scale subsidence, where it may then entrain into the MBL. The large-scale subsidence also plays a role in the strong inversions atop MBLs in the region. The MBLs transition to decoupled boundary layers toward the equator as sea surface temperatures (SSTs) increase (Wood, 2012 and references therein). The collected datasets were designed to capture this system.

P-3 latitude vs. longitude flight tracks are shown in Fig. 1 and latitude vs. altitude flight tracks are shown in Fig. 2. Sampling spanned the altitude range 70 m to 7 km, covering the MBL and the region of the LT where BBA plumes were present. Flight maneuvers included horizontal level legs, "saw-tooth" profiles through cloud layers and at plume boundaries, and vertical profiling via either ramps or square spirals. For more information on aircraft sampling strategies and maneuvers for each flight, Redemann et. al. (2021) provide several useful tables and figures. Their Table 2 and Fig. 12 explain the types of sampling flight maneuvers performed. Their tables A1a, A2, and A3 provide brief summaries of sampling activities for each P3 flight along with the number of repetitions for each flight maneuver.

Sampling took place over the three time periods: Aug. 27 – Sept. 27, 2016 (15 flights); Aug. 09 – Sept. 02, 2017 (14 flights); and Sept. 24 – Oct 25, 2018 (15 flights). Flights were typically every 2-3 days, lasted 7-9 hours, and occurred during daytime hours (7am to 5:30pm local time). Figure 2 also provides a summary of flight hours for 1 km altitude bins.

## 3. The Water Isotope System for Precipitation and Entrainment Research (WISPER)

WISPER was designed to simultaneously obtain in-situ measurements of total water and cloud water concentrations and their isotope ratios D/H and $^{18}O/^{16}O$ (schematic shown as Fig. 3). This was accomplished by having two gas-

phase isotopic analyzers (Picarro models L-2120fxi and  L-2120i), with the first capable of sampling either total water from a solid diffuser inlet (SDI, McNaughton et al., 2007) or cloud water from a counterflow virtual impactor (CVI, Noone et al., 1988; Twohy et al., 1997), and the second sampling from the SDI only. Both analyzers measure specific humidity and the isotope ratios HDO/$H_2O$ and $H_2^{18}O$/$H_2O$ (Gupta et al., 2009). From hereon out, the first water isotope analyzer is referred to as WIA1 and the second as WIA2 (Note that in the timeseries datafile documentation, WIA1 and WIA2 are referred to instead as Pic1 and Pic2, referencing the particular brand of instrument used).

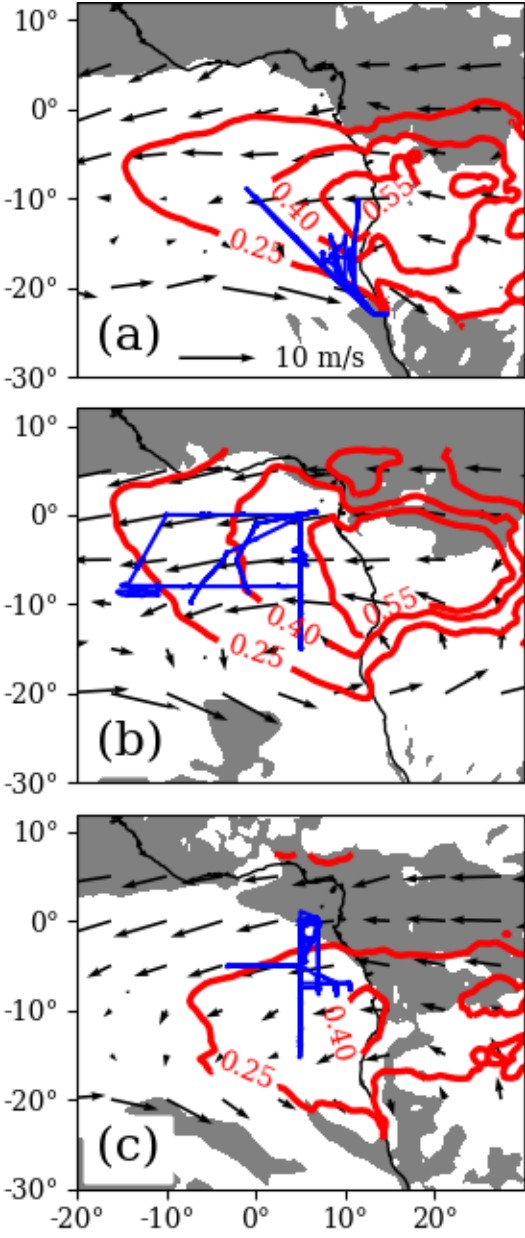

**Figure 1: ORACLES P3-Orion flight tracks (blue) for the (a) Sept. 2016, (b) Aug. 2017, and (c) Oct. 2018 sampling periods.  MERRA monthly mean aerosol optical depth (red contours) and 500 hPa winds (arrows) between latitudes 25˚S and 8˚N are shown. Shading indicates where the MERRA monthly mean 500 hPa vertical velocity is upwards, shown to emphasize that sampling took place in a region of large-scale subsidence. The boundary between shaded and non-shaded regions indicates a mean velocity of 0.**

The SDI was operated by the Hawaii Group for Environmental Aerosol Research (Howell, et al., 2020) and maintained at near isokinetic flow. Total water measurements (vapor + liquid + ice) were obtained from the SDI. For

the majority of most flights, the P-3 was not sampling cloudy air and so both WIA1 and WIA2 drew air from the SDI. Air from the inlet passed through a transfer line (labelled in Fig. 3) where a fraction of the flow was diverted to the analyzers. Flow in the SDI transfer line was generated using low pressure provided by a venturi exhaust and maintained at a flow rate of 2 SLPM with an Alicat MC-series mass flow controller (MFC). For each analyzer, a diaphragm pump was used to divert air from the transfer line to the analyzer and the flow rate is maintained by an MFC internal to the analyzer. Both diaphragm pumps were Vacuubrand MD-1 Vario models. All plumbing lines carrying sample air between the SDI and gas analyzers were 0.25 OD copper. The length of the line from the SDI to WIA1 was approximately 1.6 m and the length of line from the SDI to WIA2 was approximately 6m. Portions of the lines between the inlets and the gas analyzers as indicated in Fig. 3 were heated (using 50 ºC by Raychem brand self-regulating heat tape) to vaporize any liquid + ice water before sampling.

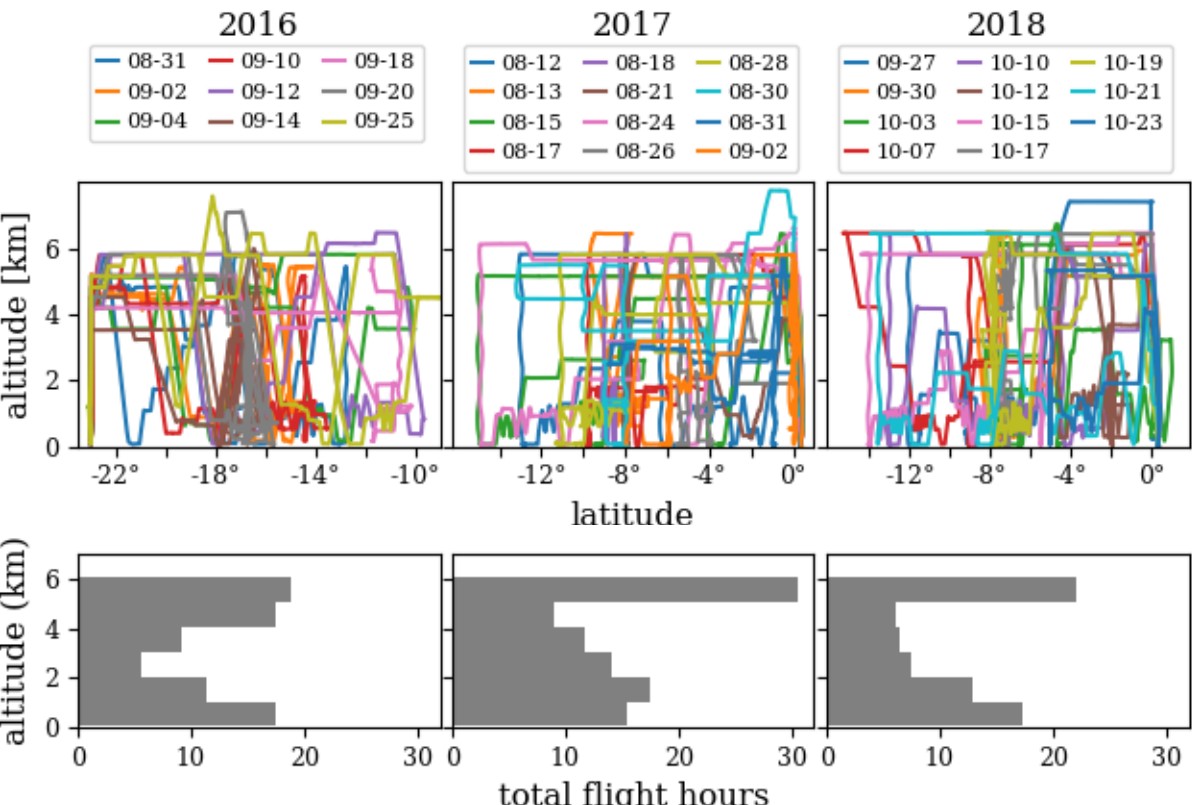

**Figure 2: P-3 latitude-altitude sampling statistics for flights where WISPER took quality data. (top row) P-3 latitude vs. altitude tracks colored by flight date ("month-day"). (bottom row): Total flight hours at each km of altitude.**

The CVI, adapted from the NSF Gulfstream-V inlet (G-V CVI), separates the condensed water particles (liquid + ice) from the ambient vapor. The CVI works by pushing dry air counterflow, opposite the direction of the inlet sampling flow, through holes in the CVI tip which prevents the ambient air from entering the inlet. The counterflow strength is tuned so that condensed water particles are able to pass via their inertia (see Noone 1988; Twohy et al., 1997). The CVI tip and transfer lines are heated so that water particles evaporate completely before sampling by the isotope analyzers. The CVI is operated at sub-isokinetic flow, leading to an enhancement of the water particle volume concentration within the transfer lines. Since the isotope analyzers have increased precision at higher water concentrations, a typical enhancement factor of 30 allows for robust measurements in clouds with liquid water content (LWC) as low as 0.04 $g/m^3$ and science-usable measurements for LWC down to 0.01 $g/m^3$. Further details on the WISPER CVI and comparison to the G-V CVI are given in Appendix A. Like the SDI, the CVI plumbing consists of a main transfer line that instruments (the WIA1 and several other ORACLES instruments) pull from. Flow along the CVI transfer line was generated by a diaphragm pump and controlled by an Alicat MC-series MFC, modified to use a wider orifice enabling a lower pressure drop for use at high altitude. Total inlet flow through the CVI line could be tuned in-flight between 2-10 SLPM, to provide a desirable cloud sampling enhancement. The dry air counterflow was dynamically adjusted to supply the sum of the bypass flow, air needed for the isotopic gas analyzer and additional flow to supply other instruments used for measuring aerosol, while maintaining the required

excess dry air counterflow. Most CVI plumbing lines were 0.25 OD copper, except for a section of 0.5 OD steel
tubing used for the first 0.5 m after air passes the fuselage wall for the 2017 sampling period. The length line from
        inlet to WIA1 was approximately 1.6 m. The CVI transfer line inside the aircraft cabin was heated to a precisely
        controlled temperature of 65 ºC with Minco model CT325 controllers.

        Picarro brand analyzers were used for joint measurements of humidity and isotope ratios (Gupta et al., 2009). For
        the 2016 sampling period there was no instrument occupying the WIA1 position. For the WIA2 position, a custom
built 5 Hz sampling instrument (Picarro model L-2120fxi) referred to as 'Mako' was used for the first three flights.
        Due to instrument issues, a 0.5 Hz instrument (Picarro model L-2120i) referred to as 'Gulper' was used for the
        reaming flights where WISPER took data. For the 2017 and 2018 sampling periods, the WIA1 position was always
        occupied by Mako. For the WIA2 position, Gulper was used for 2017 and a second 0.5 Hz instrument 'Spiny' was
        used for 2018. Table 1 shows the WISPER system status for each flight. Data were collected also during test flights,
and during transit from the Wallops Flight Facility in Virginia, via Barbados and Ascension Island, to either Walvis
        Bay in Namibia or Sao Tome. Data from these additional periods are excluded from the primary dataset and the
        presented data here.

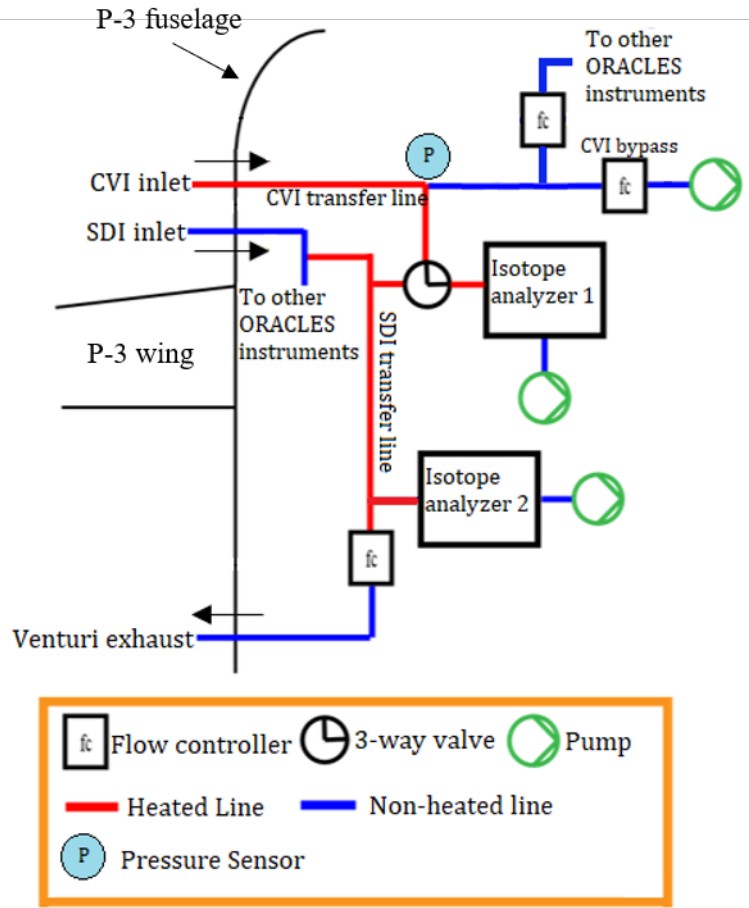

**Figure 3: Simplified schematic of the WISPER system on the P3-Orion aircraft. Arrows show direction of airflow. Both
        isotope analyzers are Picarro-brand cavity ringdown spectrometers. The first analyzer (WIA1) samples off the transfer
        lines of either a solid differ inlet (SDI) or a counterflow virtual impactor (CVI) inlet, sampling total water or cloud
        condensed water respectively. The second analyzer (WIA2) samples from the SDI only. Flow in the SDI transfer line is
        maintained by low pressure from a venturi exhaust. Flow in the CVI transfer line is maintained with a vacuum pump.**
**For each analyzer, air is pulled from the transfer line into the analyzer with a vacuum pump.**

**Table 1: WISPER status for all ORACLES flights.**

| Flight numbers | Dates | WIA1 status | WIA2 status | Notes |
|---|---|---|---|---|
| **2016** | | | | |
| RF 01-04 | Aug. 30, 31, Sept. 02, 04 | N/A | Mako, Good[1] | No known issues. |
| RF 05 | Sept. 06 | N/A | N/A | Instrument problem detected shortly after takeoff. No data available. |
| RF 06 | Sept. 08 | N/A | N/A | No data available. |
| RF 07-11 | Sept. 10, 12, 14, 18, 20 | N/A | Gulper, Good | Mako replaced with Gulper. No known issues. |
| RF 12 | Sept. 24 | N/A | Gulper, OK[2] | There are several time intervals with bad data, determined later to be associated with pressure leaks. Data for these intervals have been removed. |
| RF 13 | Sept. 25 | N/A | Gulper, Good | No known issues. |
| **2017** | | | | |
| RF 01-02 | Aug. 12, 13 | Mako, OK | Gulper, Good | Mako cal shifted from normal. Loose temperature sensor. |
| RF 03-05 | Aug. 15, 17, 18 | Mako, Good | Gulper, Good | No known issues. |
| RF 06 | Aug. 19 | N/A | N/A | Flight aborted. |
| RF 07-09 | Aug. 21, 24, 26 | Mako, Good | Gulper, Good | No known issues. |
| RF 10 | Aug. 28 | Mako, Good/OK | Gulper, Good | Some short time intervals with bad Mako readings. Bad data removed. |
| RF 11-13 | Aug. 30, 31, Sept. 02 | Mako, Good | Gulper, Good | No known issues. |
| **2018** | | | | |
| RF 01-02 | Sept. 27, 30 | Mako, Good | Spiny, Good/OK | No known issues for WIA1. WIA2 data OK *† |
| RF 03 | Oct. 02 | Mako, Bad | Spiny, Bad | No data available. |
| RF 04 | Oct. 03 | Mako, Good | Spiny, Good/OK | No known issues for WIA1. WIA2 data OK *† |
| RF 05 | Oct. 05 | Mako, Bad | Spiny, Bad[3] | No data available. |
| RF 06-13 | Oct. 07, 10, 12, 15, 17, Oct. 19, 21, 23 | Mako, Good | Spiny, Good/OK | No known issues for WIA1. WIA2 data OK *† |

[1] All measurements science usable. [2] Some bad data needed removal. [3] No data available.

* No WIA2 data at pressures below ~700 mbar.

† WIA2 cavity pressure periodically fluctuates outside normal operating range. Data removed whenever cavity fluctuated by more than 0.2 torr.


**4 Data Processing**

**4.1 Pre-processing and calibration strategy**

Over the course of the three-year experiment, three different gas analyzers were used, and each had some degree of technical challenges during each campaign. The calibration and post processing strategy were designed with several aims: 1) ensure time synchronization with other data streams, 2) consistency between the two gas analyzers that were used simultaneously in the WIA1 and WIA2 positions, 3) minimization of known instrumental measurement dependencies (specifically, with respect to dependence on humidity), and 4) ensuring that all relevant errors were

accounted for as a core aspect of the resultant datasets. This was best accomplished by aspiring to have one instrument (the highest performing one) well calibrated, and using it to transfer calibration to the other instruments. In periods when the primary instrument had failures, we were required to use a second instrument to provide absolute calibration.

Prior to any postprocessing, data were first compiled onto a common 1Hz frequency set of timestamps. Data for

Mako were binned and averaged from the raw 5 Hz to 1 Hz frequency using a boxcar weighting function. Data for Gulper and Spiny were linearly interpolated to 1 Hz from 0.5 Hz. The time series were visually inspected and any data that were clearly erroneous were removed. For the 2018 campaign, Spiny had laser-cavity pressure fluctuations outside the normal operating range once environment pressures dropped below roughly 700 mbar and those data are removed. For WIA1 CVI measurements, data where cloud water content is below 0.01 g/kg is not considered

reliable and is removed (note that with a CVI enhancement factor of typically 30, an actual cloud water content of 0.01 g/kg is measured as 0.3 g/kg by the analyzer). Isotope data are reported in "delta" notation $\delta = (R/R_s - 1)$, where R is the heavy to light molar isotope ratio, and $R_s$ the isotope ratio of the Vienna Standard Mean Ocean Water. Calibration is performed for the $\delta$ values.

**4.2 Time synchronization**

Due to the travel time of air from the sampling inlet to the gas analyzers, the WISPER measurements have a time lag between sampled air's point of inlet entry and point of measurement. To address this, a pressure-dependent time shift was applied which ultimately synchronized the data with the Passive Cavity Aerosol Spectrometer Probe (PCASP, Droplet Measurement Technologies, Rosenberg et al., 2012), chosen as the reference since its measurements are assumed to have minimal lag. Time shifts were found using a time-lag maximum cross-

correlation method ("spike matching") which relies on the instruments to measure quantities which are expected to covary. Since the isotope analyzers and the PCASP do not have clearly covarying quantities, the COMA instrument (ABB–Los Gatos Research $CO/CO_2/H_2O$ analyzer, Liu et al., 2017) was used as an intermediary. PCASP spikes in BBA are expected to align with COMA spikes in carbon monoxide, while both COMA and WISPER measure specific humidity when not in cloud. Therefore, COMA is first aligned with the PCASP, then WISPER is aligned

with COMA.

Because the WISPER plumbing was mass-flow controlled, the volumetric flow speed increases with decreasing environment air pressure and the time lag is not constant. Time corrections were formulated as a linear function of pressure. For each flight, WISPER and COMA data were separated into 50 mbar bins. For each bin, the time lag $\Delta t$ with the maximum cross-correlation between their specific humidity measurements was found. A time correction

function $\Delta t = c_1 P + c_2$ was fit to time lags vs. pressure bins using linear regression, and then applied to the data.

**4.3 WIA1 humidity and isotope ratio calibration**

Our calibration of WIA1 humidity and isotope ratios is similar in philosophy to other studies and outlined briefly here, with further details and calibration results given in Appendix B. The calibration procedure is as follows: (a) calibration of humidity against a Licor 610 Dew Point Generator (DPG), (b) correction for the humidity-dependent

bias that Picarro instruments are known to develop at lower humidites (see e.g. Schmidt et al., 2010; Tremoy et al., 2011), and (c) calibration to an absolute scale using several water standards for which the isotope ratio is known (Coplen, 1994).

WIA1 showed drift in its $\delta^{18}O$ over the three sampling periods which the absolute calibration function does not capture since it was derived before the first observation period. This was addressed with a semi-objective adjustment

term, ranging 1-3 ‰, which corrects for observed drift in $\delta^{18}O$ over the 3 years when comparing histograms of P3 data collected below 500 m in altitude (see Eqn. B3b). $\delta D$ does not show this drift. Additionally, deuterium excess ($dxs = \delta D - 8\delta^{18}O$) maxima are anomalously high in 2016 and 2017 when compared to previous studies, in proportion with the $\delta^{18}O$ drift. We speculate the origin of this shift is to at least one of two possible causes. The first

is degradation of the optical system resulting from sampling in the highly polluted biomass burning plume (BBA concentrations were higher in 2016 and 2017 then in 2018), with the design of the Picarro optical cavity excluding the possibility for mirrors to be cleaned. The second is aircraft vibrations (which were persistent and at times strong) shaking hardware to yield performance below factory specifications given for ideal lab conditions. In either case, $\delta^{18}O$ numerical values can be affected more by shifts in instrument hardware than $\delta D$ since they are ~7 x smaller in magnitude. More over, the detailed spectroscopy of each line feature differ, so they need not respond similarly. Previous studies in the Atlantic suggest that the *dxs* peak is typically observed be between 12-18 ‰ (Benetti et al., 2017), and therefore we chose *f* to bring *dxs* into this range, while being faithful to our estimates of absolute calibration. The introduction of *f* is not ideal, and fundamentally stems from limitations in the collection of calibration data under flight conditions. The manner in which this uncertainty was accounted for is discussed in the last paragraph of Appendix D.

## 4.4 WIA2 humidity and isotope ratio calibration

### 4.4.1 Cross-calibration in 2017 and 2018

For 2017 and 2018, both WIA1 and WIA2 were present in the WISPER system. WIA1 and WIA2 measured from the same inlet for the majority of each flight, except for when WIA1 was switched to the CVI in-cloud. For these years, WIA2 was cross-calibrated to WIA1. Cross-calibration was chosen over absolute calibration to ensure that the relative changes in total water and cloud water isotope ratios are as accurate as possible. The ability to compare the relative difference between these two measurements was, by design, the scientific target.

For both 2017 and 2018, the relationship between WIA1 and WIA2 specific humidity follow a line ($R^2>0.99$) that passes through the origin, with slopes of 0.9077 and 1.1007 respectively. For $\delta D$ measurements, WIA1 $\delta D$ was modeled as a polynomial of WIA2-measured $q$ and $\delta D$, and likewise for $\delta^{18}O$. Let $\delta_1$ denote a WIA1 isotope ratio measurement ($\delta D$ or $\delta^{18}O$), and $q_2$, $\delta_2$ denote WIA2 humidity and isotope-ratio measurements. The following polynomial was fit using linear regression (Python statsmodels, Seabold & Perktold, 2010):

$$\delta_1 = \sum_i^{n_q}[c_{qi} \ln(q_2)^i] + \sum_j^{n_\delta}[c_{\delta j}\delta_2^j] + \sum_k^{n_x}[c_{xk}(\ln(q_2)\,\delta_2)^k] + \varepsilon_{\delta,calib} , \tag{1}$$

where the second-to-last term is a cross-term and $c_{qi}$, $c_{\delta j}$, and $c_{xk}$ are linear fit parameters. The term $\varepsilon_{\delta,calib}$ is an error-term associated with the calibration. The orders of the polynomials $n_q$, $n_\delta$, and $n_x$ were chosen to minimize the Bayesian information criterion but capped at order 5. Polynomial fits to Eqn. 1 for both years and both isotopologues have $R^2>0.93$. Figures 4 and 5 show root-mean-squared error maps (estimating average $\varepsilon_{\delta,calib}$) for 2017 and 2018 data binned by WIA1-measured q and isotope ratios.

### 4.4.2 Calibration in 2016

For 2016, only one instrument (in the WIA2 position) was present. For the first three flights this position was filled by Mako, for which the calibration procedure is as detailed in Appendix B. For the remaining 2016 flights with available data, the position was filled by Gulper. The calibrations follow Eq. B1-B3 analogous to Mako. For Gulper, Eq. B1 $m_q = 0.909$. Parameters for Gulper $\delta(q)$ Eq. B2 are given in Table B1. For Eq. B3, a 2-point calibration was performed in the lab between the 2016 and 2017 deployments, for which the slopes (1.094 for $\delta D$ and 1.068 for $\delta^{18}O$) are trusted but not so much the offsets. Therefore, an alternate estimate of the offset is obtained by comparing histogram peaks of Gulper sub-cloud layer measurements to those of Mako measurements over similar flight tracks. This relies on the assumption that for similar synoptic conditions these two peaks should roughly coincide. Further details are given in the appendix. The uncertainties associated with this method are discussed below.

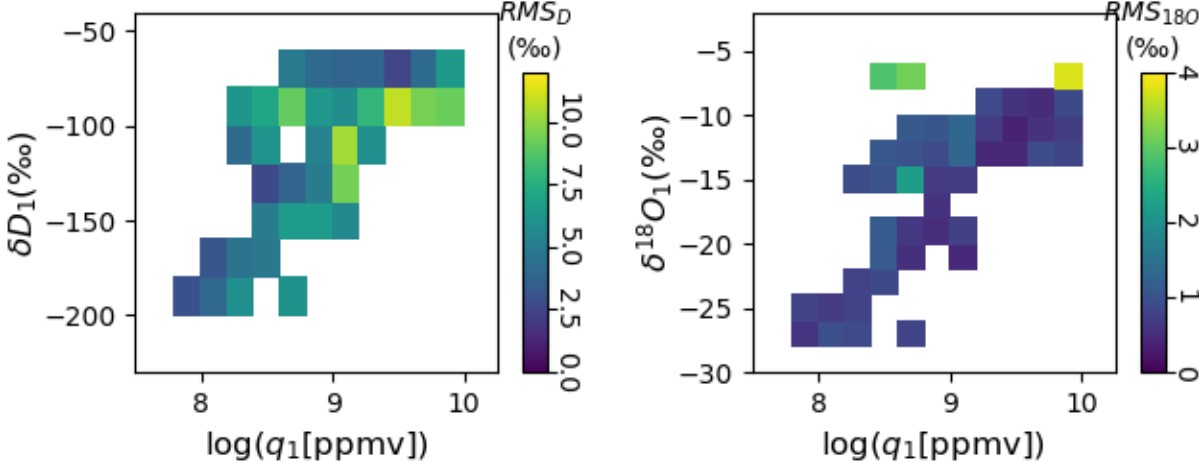

**Figure 4**: Root-mean-squared errors after WIA1-WIA2 cross-calibration for δD (left) and δ¹⁸O (right). Data are binned by WIA1-measured quantities. Horizontal axes bin width is 0.2. Vertical axis bin width is 20 ‰ for δD 2 ‰ for δ¹⁸O. Only bins for which there are at least 2 minutes of data are shown.

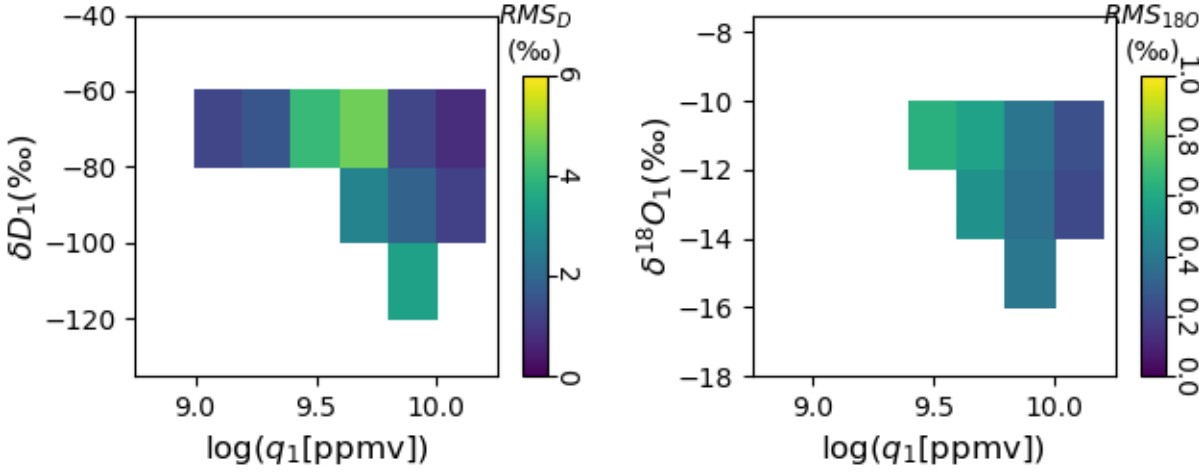

Figure 5: Same as Fig. 4 but for the 2018 sampling period, during which there was a narrower range of q and δD available for cross calibration compared to 2017.

### 4.5 Comparison of ORACLES near-surface measurements to previous studies

The data below 500 m are compared to previous studies which used ship based, near-surface measurements. Table 2 gives intervals in which 95 % of the WISPER measurements fall after averaging the 1 Hz data into 10 s blocks. Benetti et al. (2017) summarize five cruises in the Atlantic Ocean which collected water vapor isotope measurements. Leaving out the ACTIV cruise, which had exceptionally negative delta-values, the cruises find $\delta^{18}$O typically in the range (-15 ‰, -9 ‰), δD in the range (-60 ‰, -110 ‰), and (-5 ‰, 25 ‰) for *dxs* (which we inferred from their Fig. 5). These ranges agree well with our measurements. Although the 2016 and 2017 ORACLES $\delta^{18}$O measurements were purposely given a constant offset so that histogram peaks of near surface *dxs* were ~15 ‰, this does not affect the width of the $\delta^{18}$O or *dxs* intervals, which either agree with or are narrower than the cruise data. As another example, Pfahl and Sodemann 2014 Fig. 1a shows ship-based measurements of *dxs* from studies in the Mediterranean Sea (Gat et al., 2003) and Southern Ocean (Uemura et al., 2008) with ranges of  (10 ‰, 30 ‰) and (-5 ‰, 30 ‰) respectively. Again, the ORACLES mixed-layer *dxs* measurements fall within these ranges.

**Table 2: Intervals in which 95 % of the near-surface (altitude < 500 m) WISPER measurements fall after averaging the 1Hz data into 10 s blocks, given separately for each ORACLES sampling period.**

| ORACLES year | δD | δ¹⁸O | dxs |
|---|---|---|---|
| 2016 | -80.7, -61.9 | -11.8, -10.0 | 10.5, 22.1 |
| 2017 | -81.0, -64.5 | -12.6, -10.0 | 10.1, 22.4 |
| 2018 | -80.8, -70.2 | -12.0, -10.1 | 7.5, 18.9 |

## 5 Available dataset variables, illustrative examples, and uncertainties

The WISPER data are available in three formats. A quick understanding of the spatial trends over the three sampling periods can be obtained using the mean latitude-altitude curtains. Individual vertical profiles at 50 m resolution have been isolated as a second dataset, with 219 total profiles over the three sampling periods. Lastly, the 1 Hz timeseries for each flight, merged with other ORACLES variables, are also available.

### 5.1 Latitude-altitude curtains

Mean curtains of water concentration and total water isotope ratios $\delta D$ and $\delta^{18}O$ (figures 6-8; $dxs = \delta D - 8\delta^{18}O$ is plotted instead of $\delta^{18}O$) are available for each sampling period as NetCDF files on Zenodo. WIA1 measurements are used where available and filled with WIA2 measurements where unavailable (e.g. when WIA1 was on the CVI). The isotope ratios are weighted by water concentration. Curtains were generated by first averaging flight data into 30 s blocks and then averaging onto 100 km altitude by 0.2˚ latitude grids using Gaussian kernel density estimation (KDE), with bandwidth estimated by Silverman's rule of thumb. The files include standard deviations at each grid point, also computed via KDE, as well as the weighted number of samples used for the KDE calculation at each grid point.

Mean 180 ppbv carbon monoxide (CO) contours and estimates of MBL tops are included in figures 6-8 for a brief plausibility check of the WISPER measurements for scientific use but are not included in the curtain dataset. The CO contours bound the 2 – 5 km region where biomass burning plumes were often present. Within the MBL, $\delta D$ values are typically larger and $\delta D$ variation smaller than in the lower free troposphere (LFT), reflecting connection to the ocean surface. In comparison, $dxs$ ranges are similar in the MBL and LFT except for some 2016 regions where mean $q$ becomes lower than about 2 g/kg. This would be the case if most phase change processes in the LFT are dominated by near equilibrium fractionation.

Above the MBL, $q$ and $\delta D$ relationships vary. For the 2016 sampling period, there is visually clear spatial correlation between $q$ and $\delta D$, which may be in part due to vertical structure in water concentration – a moist MBL is topped by a dry air 'wedge' with a moist layer further up. The coincidence of increased $q$ and $\delta D$ with the CO contour supports the idea that this 'plume' of moisture and isotopes came from the African planetary boundary layer (PBL), since CO is a good indicator of the biomass burning plumes targeted during ORACLES (see e.g. Zuidema et al., 2016; Redemann et al., 2021). Airmass origin may explain some of the LFT signal for the other two sampling periods. For example, in 2017 some LFT features in $q$ and $\delta D$ match (e.g. the dark orange $\delta D$ feature at 3 km centered at 6 ˚S while other do not (e.g. dark orange feature at 2 km centered at 13 ˚S). For 2018, the highest LFT $\delta D$ values reside within the CO contour even though $q$ is higher toward the equator at almost all altitudes.

### 5.2 Vertical profiles

The WISPER data (total water and cloud water quantities) merged with latitude, longitude, altitude, temperature, and pressure are available for individual vertical profiles, collected into NetCDF files on Zenodo. Total water quantities use primarily WIA1 measurements and are filled with WIA2 wherever WIA1 is unavailable. Vertical profiles here have been defined as any P-3 flight sequence where the aircraft had an overall ascending or descending trajectory covering the altitude range 70 m to 7 km and typically within two hours. For this reason, the dataset includes both deliberate vertical profiling (e.g. a constant 1500 ft/min vertical speed which completes the 7 km profile in around 15 minutes), as well as profiles interspersed with horizontal or saw-tooth pattern sampling at altitudes of interest. In either case, the data for each profile is averaged into 50 m vertical bins resulting in a 1-1 function of WISPER variables with height. Time bounds are provided as a variable in the dataset for users looking to isolate only those profiles performed within a shorter time interval. In total there are 84 profiles for the 2016 sampling period, 79 profiles for 2017, and 56 profiles for 2018.

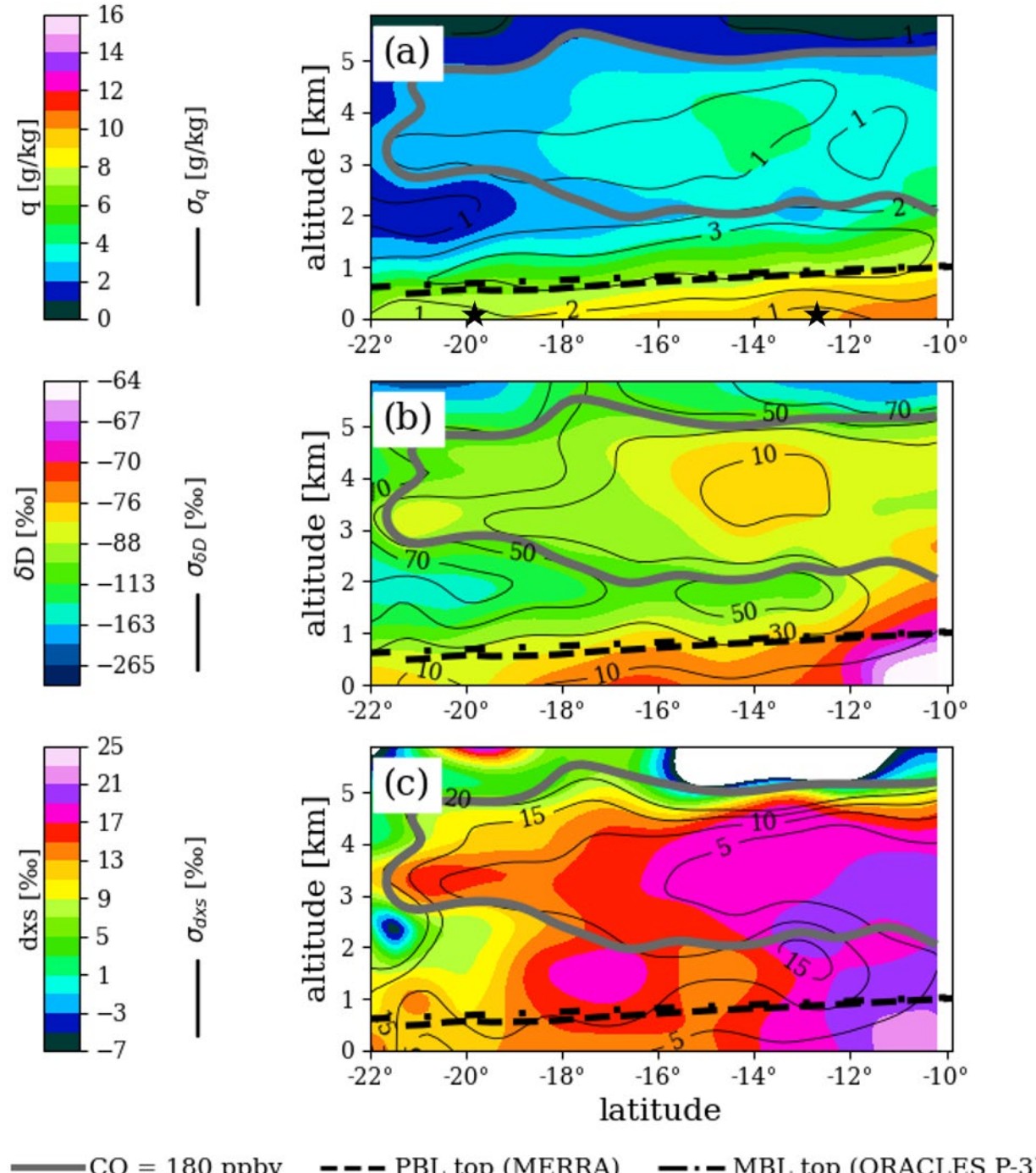

Figure 6: Latitude-altitude curtains of WISPER mean $q$ (a), $q$-weighted δD (b), and $q$-weighted *dxs* (c) for the ORACLES 2016 sampling period. Curtains were generated using a gaussian kernel estimation method after averaging the 1 Hz data into 30 s blocks and removing any data where $q < 0.2$ g/kg. Thin black contours denote standard deviations. Thick grey contour shows 180 ppbv in-situ carbon monoxide measured by the COMA system; it bounds the 2 – 5 km region where biomass burning air was often present. Black dashed line shows planetary boundary layer top taken from MERRA monthly mean output for Sept. 2016. Black dash-dotted line shows a linear regression of MBL capping inversion bottom vs. latitude using P-3 in-situ measurements. Inversion bottoms were estimated using vertical profiles of temperature and relative humidity. Black stars at the bottom of (a) are placed at the latitudes of the two vertical profiles in Fig. 9.

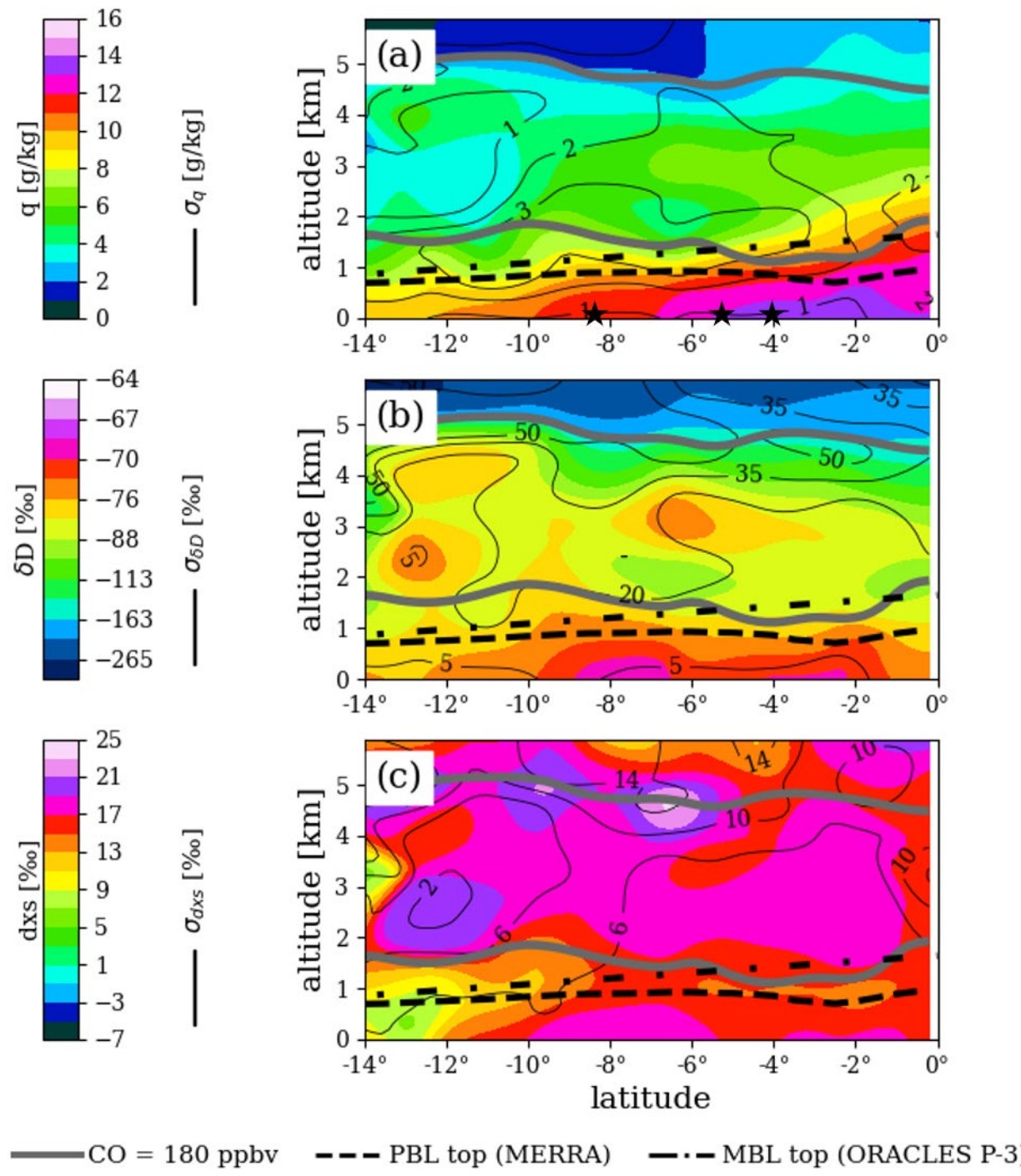

340

**Figure 7: Same as Fig. 6 but for the ORACLES 2017 sampling period. Black stars at the bottom of (a) are placed at the latitudes of the three vertical profiles in Fig. 10.**

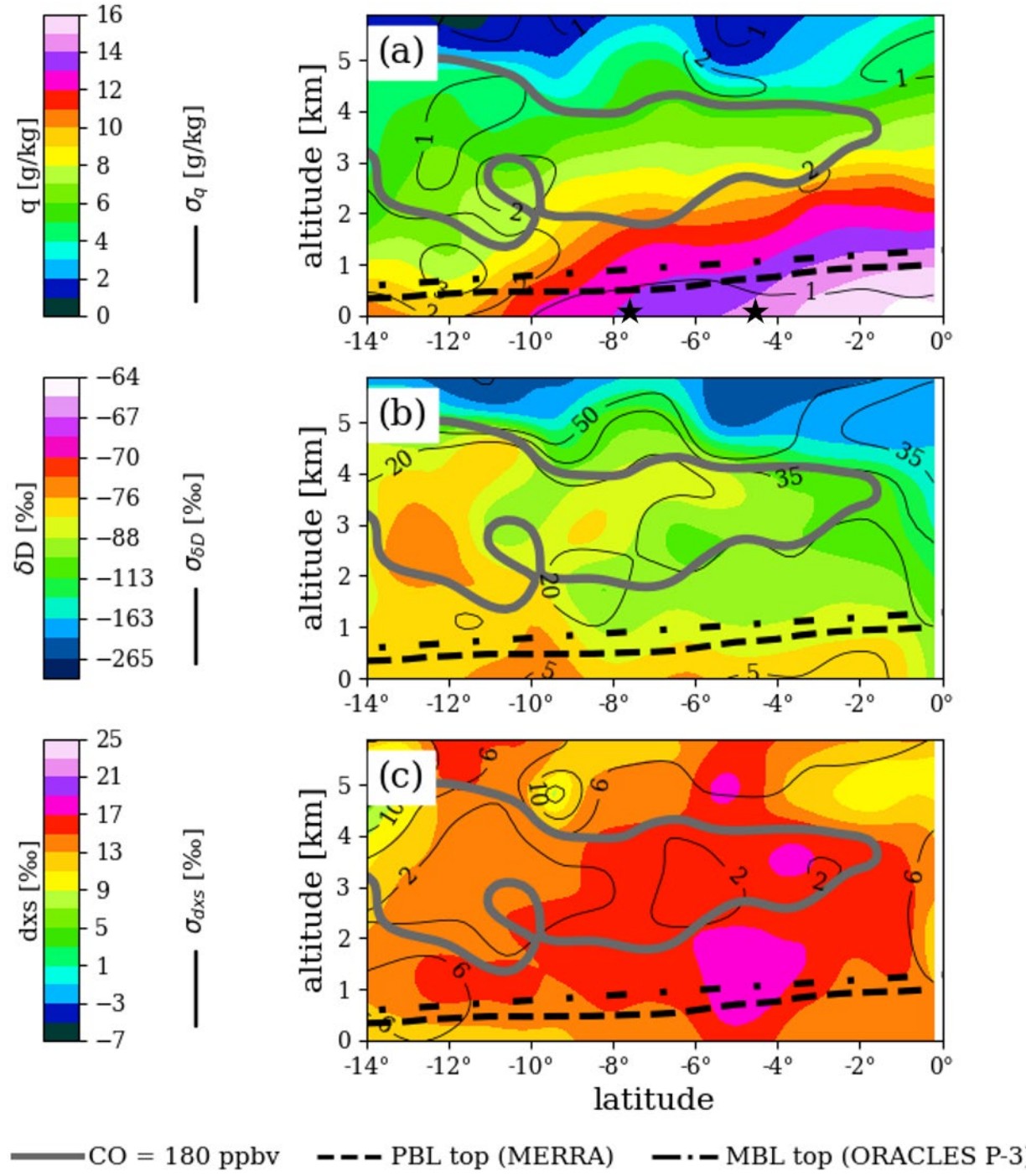

345

**Figure 8: Same as Fig. 6 but for the ORACLES 2018 data. Black stars at the bottom of (a) are placed at the latitudes of the two vertical profiles in Fig. 11.**

350

Figures 9-11 show examples of individual vertical profiles WISPER total water measurements (concentration and isotope ratios) from each sampling period. Profiles of potential temperature and CO are included as a plausibility check. Sharp gradients in potential temperature between 500 – 1200 m indicate MBL top. Below MBL, δD is comparatively uniform in comparison to its fluctuations in the LFT, in line with surface-coupling. For the 2016 profiles, the presence of a dry, isotopically depleted layer just above MBL with a plume of moisture further up is evident and agrees with peaks in CO. Across observation periods, δD in the LFT appears to covary with CO more so than $q$ (e.g. particularly clear in Fig. 11a), supporting the idea that δD can indicate the presence of African PBL air. The agreement between vertical features in δD and CO, down to ~200 m in some cases, also supports the precision of the WISPER measurements. *dxs* is harder to interpret; one trend common in most (but not all) the profiles is the *dxs* is higher in the LFT plumes than at the surface by 3 – 6 ‰. Uncertainties of ~2.5 ‰ and ~0.5 ‰ for δD and δ$^{18}$O respectively (see Section 5.4) result in a *dxs* uncertainty of 2.9 ‰, making some of the observed trends possible but borderline to resolve. However, Fig. 10b presents a compelling case for future study, where *dxs* jumps from 13 ‰ near the surface to 4 ‰ in the decoupled layer, and then up to 20 ‰ in the overlying LFT.

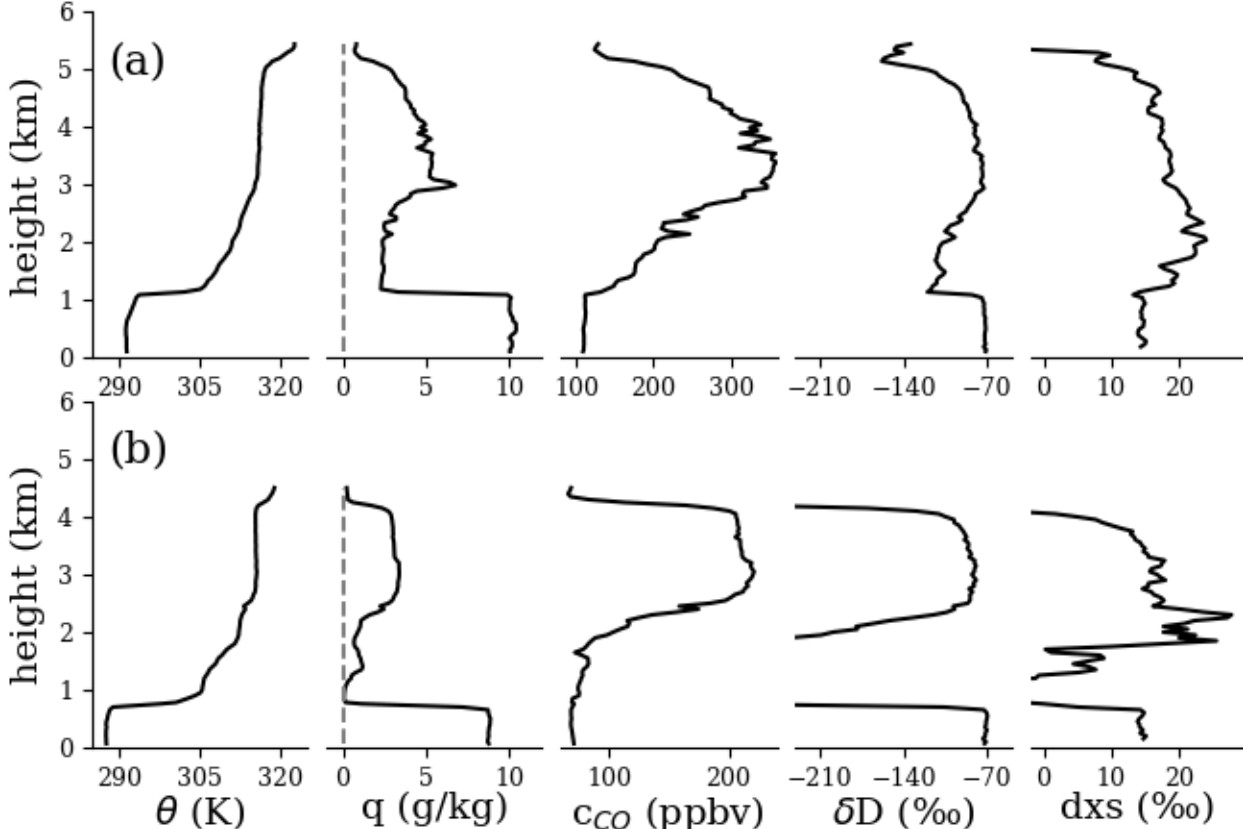

**Figure 9: Examples of individual vertical profiles for the 2016 sampling period. Potential temperature, humidity, CO concentration, total water δD, and total water dxs for profiles taken on (a) Aug. 31 (mean latitude 12.75 ˚S, mean longitude 2.55 ˚E, mean time 11:42 UTC) and on (b) Sept. 14 (19.9 ˚S, 10.2 ˚E, 09:09 UTC). Data were averaged into 50 m vertical bins. The *dxs* has an additional 3-bin running mean applied. The δD and dxs in (b) become very low where $q$ approaches 0 and are not shown.**

Most of the profiles in figures 10 and 11 show a decreased δD region above MBL top (highlighted) in comparison to the air both above and below. Such a signal is consistent with expectations for a clean, dry air wedge as seen clearly in the 2016 curtain and profiles, but is more subtle in the 2017 and 2018 profiles. Alternatively, the highlighted regions could have experienced precipitation, which preferentially removes heavy isotopes in comparison to mixing processes. In the exceptionally depleted region in Fig. 11b, rain re-evaporation may further play a role (see e.g. Noone 2011). The data set captures these types of features with high fidelity enabling these hypotheses to be examined. These signals were determined to not be measurement artifacts such as memory effects from ascending

vs. descending aircraft trajectories. While memory effects were observed, they occurred only for transitions to very dry conditions (~< 1 g/kg) over an altitude range of 100 – 200 m.

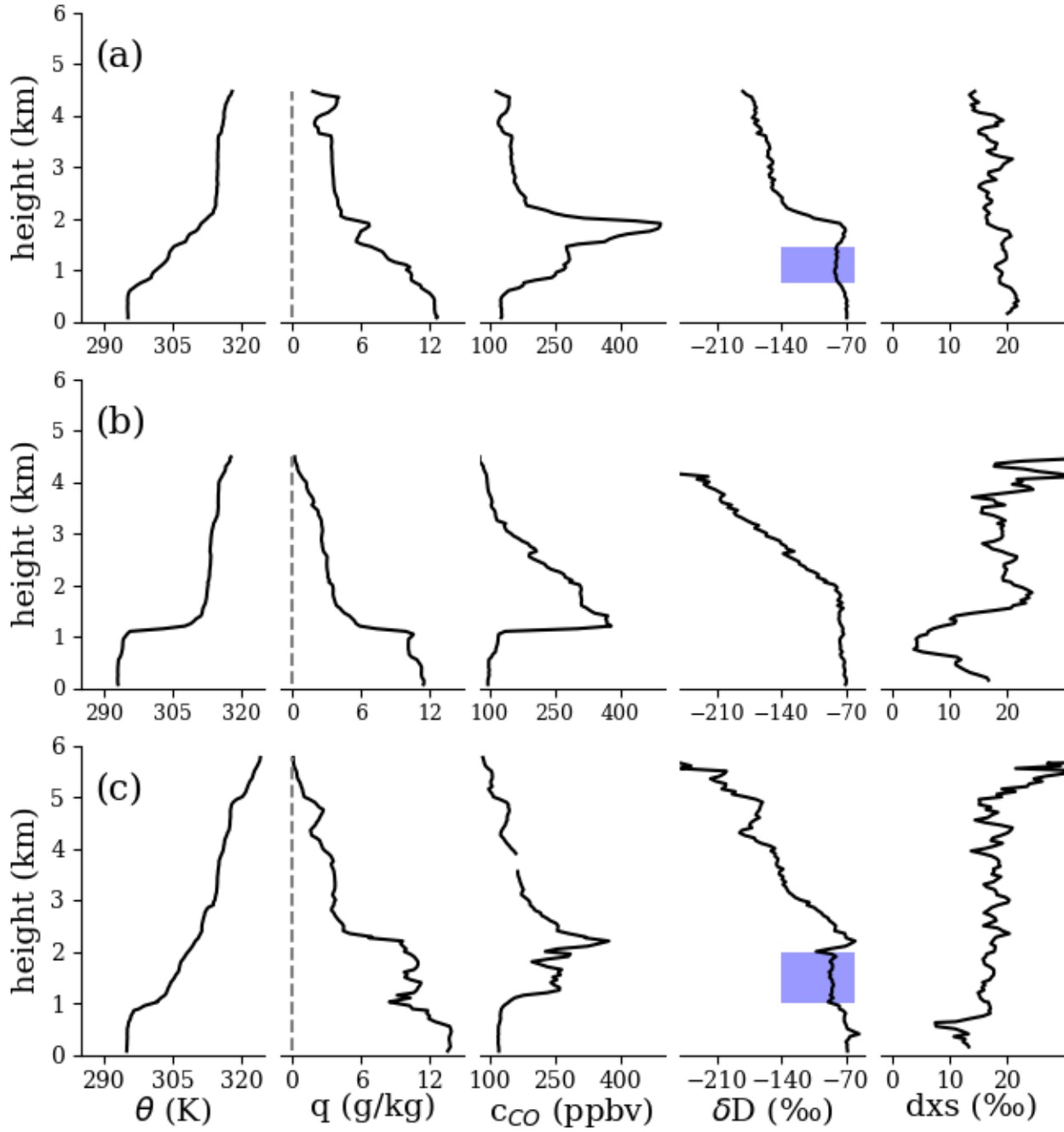

380

**Figure 10: Similar to Fig. 9 but for the 2017 sampling period. (a, b): Two vertical profiles taken on Aug. 15: (a) mean latitude 4 ˚S, mean longitude 4.95 ˚E, mean time 15:16 UTC; (b) 8.4 ˚S, 5.0 ˚E, 13:50 UTC. (c) One profile from Aug. 26 (5.2 ˚S, 5.0 ˚E, 11:17 UTC). Blue highlighted regions are discussed in the main text.**

385

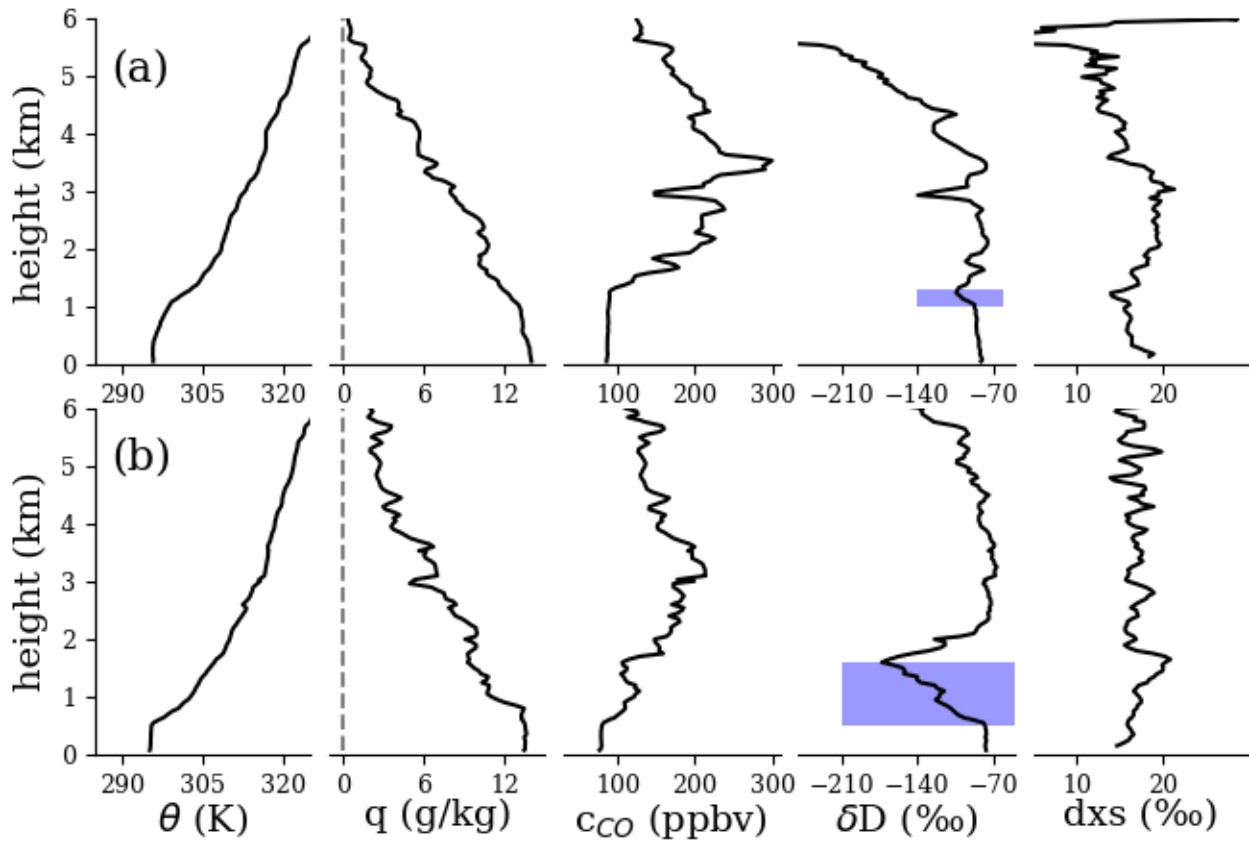

**Figure 11: Similar to Fig. 9 but for the 2018 sampling period. Vertical profiles taken on (a) Oct. 03 (mean latitude 4.6 °S, mean longitude 5.05 °E, mean time 13:10 UTC) and (b) Oct. 19 (7.8 °S, 9.0 °E, 10:30 UTC). Blue highlighted regions are discussed in the main text.**

Lastly, Fig. 12 provides a comparison of water vapor and total water quantities capable with this dataset. Water vapor $\delta$D and *dxs* were derived using $q$, LWC, and total water measurements of $\delta$D and $\delta^{18}$O. For single vertical profiles, the $\delta$D of the liquid is higher than the vapor, reaching values greater than or equal to 0 ‰. This would be the case for equilibrium fractionation in the cloud environment where temperatures are lower than that at the surface where the vapor was produced. With the uncertainties in WIA1 and WIA2 measurements given Section 5.4, the observed fractionation cannot be distinguished from equilibrium fractionation except for the tops and bottoms of the profile. This holds for *dxs* as well, which should have close to no change if equilibrium fractionation is occurring. Summarizing all profile data into probability distributions (PDFs) of observed fractionation, defined as

$$\alpha_{obs} \equiv \frac{R_l}{R_v} = \frac{\delta_l + 1000}{\delta_v + 1000} \tag{2}$$

(subscripts $l$ and $v$ for liquid and vapor respectively), Fig. 12c, d shows that the observed fractionation PDFs have a primary peak near equilibrium. Secondary peaks occur near $\alpha_{obs}=1$ for both $\delta$D and $\delta^{18}$O. More investigation is needed to determine whether this is a measurement artifact or true signal. If the latter, it would imply no fractionation, which seems unreasonable for condensation processes. This leaves some unique droplet evaporation process as the most likely candidate.

405

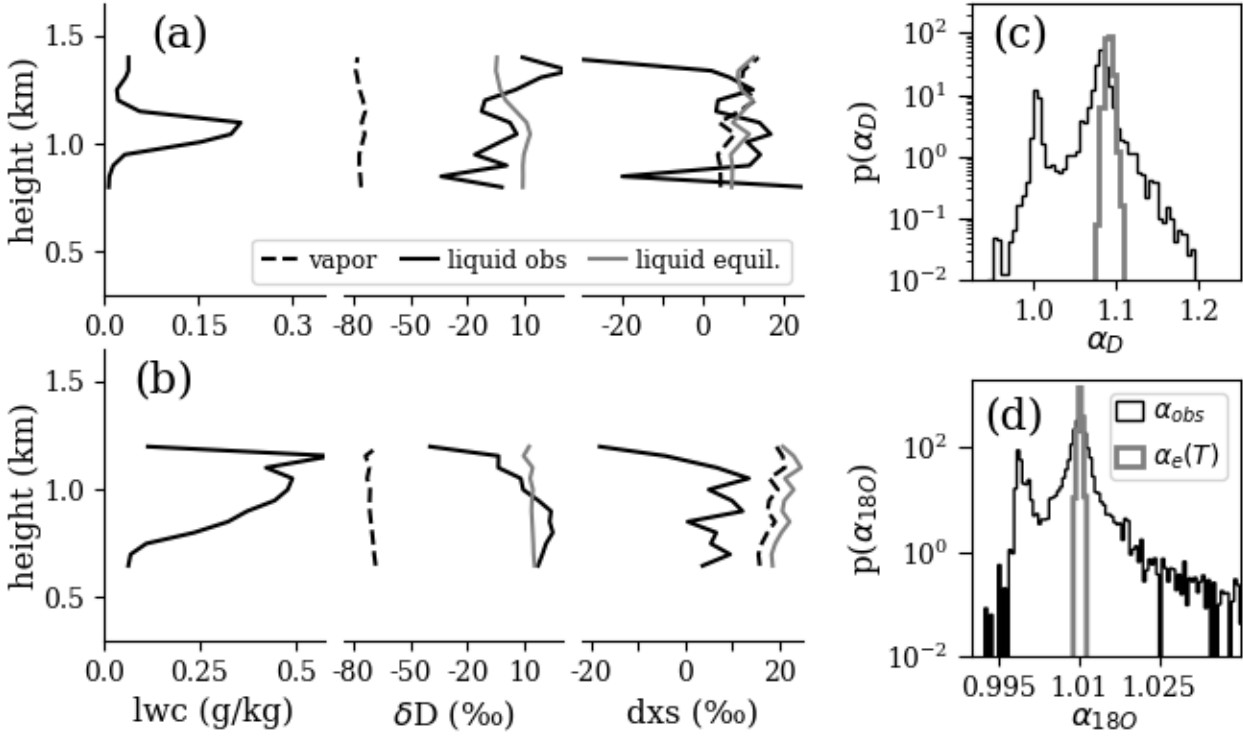

**Figure 12: (a, b): Examples of individual vertical profiles for lwc and isotope ratios of vapor and cloud liquid. Profiles predicted by equilibrium fractionation using temperature measurements are included. (c, d) Probability distributions of observed fractionation factors, defined as in Eq. 2, for all profile data below 4 km (taken roughly as the freezing level), and equilibrium fractionation factors computed from temperature.**

### 5.3 Time series

Time series for each flight are available as NetCDF files with one file per flight. For these files the WISPER measurements have been merged with most other ORACLES variables and the P-3 spatial data (longitude, latitude, altitude). For this reason they are referred to in the online directory as 'merged' files. They are available at 1 Hz frequency (the common timestep all ORACLES instruments interpolated to). The directory includes 0.1 Hz files but these should be ignored as they do not contain the most recent version of WISPER data or those of other instruments. Figure 13 shows a brief schematic to navigate to these files from the DOI homepages. Table 3 lists the WISPER variable names and descriptions included in the merged files. The WISPER data alone (e.g. water concentrations, isotope ratios, and CVI quantities) are also available as smaller files but do not include P-3 spatial data. The remainder of this section covers best usage practice for the time series data.

Unless the user is familiar with Picarro measurements or has consulted the authors, it is recommended that the data be averaged to at least 10 second averages (0.1Hz frequency), at which point each measurement can be considered independent. The 1Hz data is robust scientific analysis, but care is needed to avoid possible autocorrelation that depends on environmental factors. It is also recommended that SDI data collected at humidity < 0.5 g/kg are excluded from primary analyses. Note that for CVI data with an enhancement factor of ~30, measured water concentrations of 0.5 g/kg correspond to an actual cloud water content of 0.017 g/kg. WIA1 data (variables ending in either "_tot1" or "_cld" in the datafiles, standing for total water and cloud water quantities respectively) should be used wherever available since the 5 Hz instrument which occupied the WIA1 position has a faster sampling time and resolves smaller features. Additionally, both the calibration and uncertainties for WIA1 are better known than for WIA2, and therefore the WIA1 data should be used as a preference. The 10 second average during typical ascents and descents of 1000 ft/minute corresponds to approximately 50 meter vertical resolution for profiles. Many of the in-cloud profiles were performed at 500 ft/second, for improved vertical resolution. The WIA2 data is most useful for times where WIA1 is on the CVI and one wants in-cloud total water measurements to compare to the condensed cloud water measurements. The only exception to the above is for the flights on August 12th and 13th, 2017. WIA1

experienced problems on those days and it is suspected its calibration was altered. Therefore it is recommended that WIA2 data is used for those two days. Only the flights listed in Table 1 are applicable to most users. Data for transit flights to and from the study regions are placed in the directory but have been minimally processed.

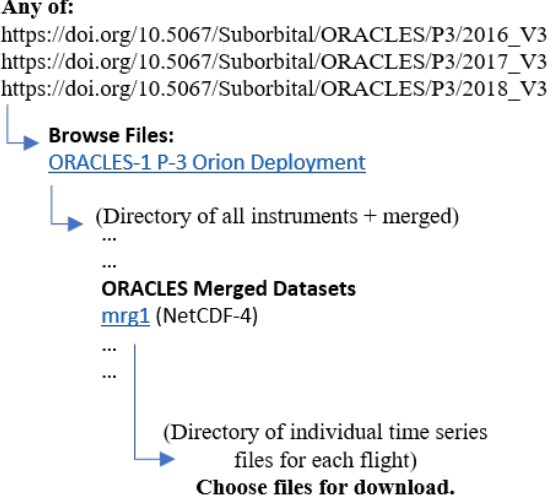

440

**Figure 13: Schematic showing user navigation from the DOI homepages to the merged time series files (containing WISPER data along with other ORACLES variables).**

**Table 3: WISPER variables and descriptions in the merged timeseries files. Note that in the datafiles, the first and second isotope analyzers are referred to as 'Pic1' and 'Pic2', reflecting the Picarro-brand used, whereas in this paper they are**
445 **referred to by 'WIA1' and 'WIA2'. Descriptions of the analyzers can be found in Section 3. Variables ending in '_tot1' and 'cld' are only available for the 2017 and 2018 sampling periods.**

| Variable name | Units | Data file description |
|---|---|---|
| wisper_valve_state | 0/1 | 1 if Pic1 is on CVI, 0 if Pic1 is on SDI. |
| h2o_tot1 | ppmv | Total water mixing ratio measured with Pic1. |
| h2o_tot2 | ppmv | Total water mixing ratio measured with Pic2. |
| h2o_cld | ppmv | Cloud water mixing ratio (including ice). |
| dD_tot1 | ‰ | Total water D/H ratio (delta-notation) measured with Pic1. |
| std_dD_tot1 | ‰ | 1 Hz precision in dD_tot1. |
| dD_tot2 | ‰ | Total water D/H ratio (delta-notation) measured with Pic2. |
| std_dD_tot2 | ‰ | 1 Hz precision in dD_tot2. |
| dD_cld | ‰ | Cloud water D/H ratio (delta-notation). |
| std_dD_cld | ‰ | 1 Hz precision in dD_cld. |
| d18O_tot1 | ‰ | Total water $O^{18}/O^{16}$ ratio (delta-notation) measured with Pic1. |
| std_d18O_tot1 | ‰ | 1 Hz precision in d18O_tot1. |
| d18O_tot2 | ‰ | Total water $O^{18}/O^{16}$ ratio (delta-notation) measured with Pic2. |
| std_d18O_tot2 | ‰ | 1 Hz precision in d18O_tot2. |
| d18O_cld | ‰ | Cloud water $O^{18}/O^{16}$ ratio (delta-notation) measured with Pic1. |
| std_d18O_cld | ‰ | 1 Hz precision in d18O_cld. |
| cvi_lwc | g/kg | Cloud liquid+ice water content, derived from 'h2o_cld'. |
| cvi_enhance | none | Enhancement factor for the CVI inlet. |
| cvi_dcut50 | μm | Cutoff diameter for CVI inlet. |
| cvi_inFlow | SLPM | Intake flow of CVI inlet. |
| cvi_xsFlow | SLPM | Excess dry air counterflow for CVI inlet. |
| cvi_userFlow | SLPM | Sum of CVI flow rates for all other instruments. |

## 5.4 Uncertainties

Uncertainties were estimated using Monte Carlo methods and then fitting a polynomial equation to errors as a function of q and the respective δ-quantity. The details of the uncertainty estimation as well as the polynomial and parameter fit values are given in Appendix D. However, for most studies the values quoted in tables 4a (WIA1) and 4b (WIA2) will suffice. For users of the latitude-altitude curtain data, only the values in Table 4a are needed. They are given for four typical situations experienced in ORACLES: high humidity MBLs, moderate humidity in the FT (typically when sampling BB-plumes), low humidity FT, and very low humidity FT.

**Table 4a: Characteristic WIA1 isotope ratio errors (e.g. +/-2σ gives 95% confidence intervals) for four typical situations observed in ORACLES. These include MBL sampling and FT sampling at a few humidities (the q and δ values under each header are typical values observed in that situation). Additionally, values for two separate use cases are included: studies looking at relative trends in the ORACLES WISPER dataset vs. comparison of this dataset to others or to theory.**

| use case | MBL q ~ 13 g/kg δD ~ -70‰ δ$^{18}$O ~ -10‰ | | Moderate humidity FT q ~ 4 g/kg δD ~ -100‰ δ$^{18}$O ~ -14‰ | | Low humidity FT q ~ 1.8 g/kg δD ~ -150‰ δ$^{18}$O ~ -20‰ | | Very low humidity FT q ~ 1 g/kg δD ~ -250‰ δ$^{18}$O ~ -34‰ | |
|---|---|---|---|---|---|---|---|---|
| | σ$_D$ | σ$_{18O}$ | σ$_D$ | σ$_{18O}$ | σ$_D$ | σ$_{18O}$ | σ$_D$ | σ$_{18O}$ |
| 0.1Hz data; Relative trends in WIA1 measurements, not requiring an absolute scale. | 1.8‰ | 0.4‰ | 2.7‰ | 0.5‰ | 4.5‰ | 0.7‰ | 10‰ | 1.3‰ |
| 0.1Hz data; comparison of WIA1 to other datasets or to theory. | 4.3‰ | 1.2‰ | 5.5‰ | 1.3‰ | 6.5‰ | 1.3‰ | 11‰ | 2.0‰ |

**Table 4b: Same as Table 4a but for WIA2 measurements.**

| use case | MBL q ~ 13 g/kg δD ~ -70‰ δ$^{18}$O ~ -10‰ | | Moderate humidity FT q ~ 4 g/kg δD ~ -100‰ δ$^{18}$O ~ -14‰ | | Low humidity FT q ~ 1.8 g/kg δD ~ -150‰ δ$^{18}$O ~ -20‰ | | Very low humidity FT q ~ 1 g/kg δD ~ -250‰ δ$^{18}$O ~ -34‰ | |
|---|---|---|---|---|---|---|---|---|
| | σ$_D$ | σ$_{18O}$ | σ$_D$ | σ$_{18O}$ | σ$_D$ | σ$_{18O}$ | σ$_D$ | σ$_{18O}$ |
| 0.1Hz data; comparisons of WIA2 to WIA1 measurements. | 5.6‰ | 0.6‰ | 6.0‰ | 0.6‰ | 7.4‰ | 0.9‰ | 13.8‰ | 1.5‰ |
| 0.1Hz data; comparison of WIA2 to other datasets or to theory. | 6.8‰ | 1.3‰ | 7.7‰ | 1.4‰ | 8.8‰ | 1.4‰ | 15.1‰ | 2.2‰ |

## 6 Final remarks

In-situ measurements of total water and cloud water concentrations and corresponding heavy isotope ratios δD and δ$^{18}$O were made via the WISPER system aboard the P-3 Orion aircraft during the NASA ORACLES project. These measurements were made alongside a wide range of other meteorological, trace gas, and aerosol variables. The project entailed measurements in the southeast Atlantic marine boundary layer and lower troposphere over latitudes 22˚S to the equator, and over the months of Sept. 2016, Aug. 2017, and Oct. 2018. WISPER successfully collected data for thirty-four research flights, which were evenly spread over the three months and collected over 300 hours of

1 Hz data. Due to ample data at all levels in the LT spanning 70m to 7km, this dataset provides valuable information on the vertical structure of isotopic compositions in the region.

In this paper, the study region and sampling strategy have been presented, followed by an overview of the WISPER system and calibration methods. The three data formats available to the users are each covered (latitude-altitude curtains, individual vertical profiles, and timeseries), with illustrative examples to highlight some features of the dataset and provide a plausibility check. The WISPER measurement system was designed to deliver paired condensed phase isotope ratio information with total water by using two inlets. The calibration and demonstration of LWC and cloud water isotope ratios is a unique feature of the dataset described herein. Further, the latitude-altitude curtains are the most comprehensive compilation of lower tropospheric in-situ profile measurements presently available. For the vertical profiles, the fact that structure in $\delta$D and *dxs* are captured encourages their use to constrain vertically resolved models that include isotopes. There are now idealized, large eddy simulation, and GCM models which include isotopes (e.g., Galewsky et al., 2016 and references therein) and the contribution of convection and cloud microphysics to the vertical structure of isotopic content can be compared to vertical profiles from the ORACLES WISPER measurements. Further, traditional thermodynamic analyses such as *q* vs. $\theta_e$ mixing diagrams (e.g. Betts and Albrecht, 1987) to diagnose the vertical structure of MBLs could be supplemented by similar isotope ratio thermodynamic charts. Utilizing the ORACLES WISPER dataset with these types of analytical approaches can refine our conceptual understanding of MBL energy and moisture budgets. Further, comprehensive modeling which includes both isotopes and other variables such as temperature could provide stronger quantitative constraints on these budgets.

**Appendix A: The WISPER CVI**

The CVI inlet used in this study was adapted from the NSF Gulfstream-V inlet (G-V CVI) deployed frequently over the past several decades. All of the hardware mounted to the aircraft fuselage was identical to the G-V CVI, with some important changes to the heaters to reduce the possibility of cold spots occurring along the sample line where water might condense and reevaporate. This is critical for accurate measurements of isotopologues of water. A new rack-mounted heater and flow control electronics unit was designed and built with contemporary components, in part because many components used in the two-decade-old G-V CVI electronics unit are obsolete, but also to improve several aspects of the counterflow operation critical for mitigating issues that might impact isotope ratio measurements. In particular, the relatively slow Omega multi-channel heater controller used on the G-V CVI was replaced with four separate, fast heater controllers (Minco CT-325) for more stable temperature control. Whereas the platinum RTD temperature sensor used for sensing temperature on the ~50-cm long, 2.5 cm O.D. stainless steel sample line that extends from probe tip to the mounting plate of the inlet on the G-V is located at the extreme downstream end of the sample line (i.e., at the mounting plate), the Pt-RTD was placed approximately 15-cm downstream of the junction with the probe tip to provide more uniform heating in the critical droplet-evaporation region. Finally, a new, independent heating zone was added to the P-3 pylon necessary to extend the probe tip beyond the aircraft boundary layer, whereas the G-V CVI system uses a single heating zone for the transfer line when the inlet is attached to a pylon (e.g., when mounted to the NCAR C-130 aircraft).

New, fast-response flow controllers were used for more precise control of counterflow. In addition, rather than using a flow controller for each individual instrument sampling from the CVI, as is currently the arrangement on the G-V CVI, two mass flow controllers (MFCs) were used to maintain a more stable sample flow. One (Alicat model MC-series, adjusted to use a wider orifice to enable a lower pressure drop at higher altitude) served to maintain a mass flow ranging from 2-5 STP liters per minute (SLM) as a bypass to the first isotopic analyzer (Fig. 3). The second was a high-flow MFC (Alicat model MCW, 20 SLPM) placed after the CVI transfer line split-off to the other instruments using the CVI. This MFC provided larger flow of 5-9 STP SLPM necessary to reduce the enhancement factor in clouds with high cloud-water contents and reduce the risk of condensation of water in the sample lines. The flow through the high-flow MFC was also used as a source for other instruments measuring from the CVI inlet (as described elsewhere).

New C++ software was also developed for more precise control of the feedback loop necessary to maintain an excess counterflow and limit infiltration of ambient air. This feedback loop operated at 10 Hz, as opposed to 1 Hz on the G-V CVI, reducing the impact of oscillations that allow "leakage" of ambient air into the inlet during periods of strong turbulence.

## Appendix B: WIA1 humidity and isotope ratio calibration

### B.1 Humidity calibration

Picarro gas analyzers report water abundance proportional to specific humidity (i.e., "wet" mixing ratio measured as the ratio of vapor pressure to total air pressure), fundamentally determined from infrared absorption by $H_2O$ relative to optical cavity pressure (Gupta et al., 2009). Specific humidity calibrations were calibrated using a Licor 610 Dew Point Generator (DPG). The DPG was used to produce saturated air at preset (dew point) temperatures between 500 ppmv (0.31 g/kg) and 20,000 ppmv (12.5 g/kg), which was then sampled by the gas analyzer. The relationship between Picarro-measured humidity $q_{pic}$ and DPG-known humidity $q_{DPG}$, was robustly linear ($R^2>0.98$), with the same slope use for all ORACLES years:

$$q_{DPG} = m_q * q_{pic} , \tag{B1}$$

where the water concentrations are in units of ppmv. For Mako, $m_q = 0.851$.

### B.2 Correction for humidity dependence of isotope ratios

Picarro isotope ratio measurements both develop a bias and become less precise as humidity decreases (see e.g. Schmidt et al., 2010; Tremoy et al., 2011). This bias was quantified by using the Picarro to measure air of constant isotope ratios ($\delta D$ and $\delta^{18}O$) diluted with a progressively higher fraction of ultra-grade dry air (Airgas product AI UZ300, specified as less than 2 ppmv $H_2O$). Each dilution is sampled for 3-7 min (as needed to get adequate statistics) and the mean is taken. The deviation of both mean $\delta D$ and $\delta^{18}O$ from those measured at the highest humidity (~18,000ppmv, or ~11g/kg) was fit with the function:

$$\Delta\delta_i(q) = a_i * \left[ ln\left( \frac{50,000}{q[ppmv]} \right) \right]^{b_i} , \tag{B2}$$

where $q$ is the measured humidity in ppmv, $a$ and $b$ are fit parameters, subscript $i$ is for the isotope species (D or 18O) and 50,000 ppmv is chosen as an asymptotically high humidity. Equation (B2) was fit to calibration data separately for each ORACLES year using non-linear least squares regression (Python SciPy's optimize package, Virtanen et al., 2020). Table B1 summarizes calibration data taken for each year and the fit parameters obtained, and Fig. B1 plots the corrections. The calibration parameters do not change significantly if the 50,000 ppmv term in the natural log is replaced with 18,000 ppmv. The calibration for 2017 is different than the other years. This alteration is attributed to a loose thermistor during the 2017 deployment which was temporarily fixed in the field and then permanently reattached before the 2018 deployment.

**Table B1: Parameter values for Equation (B2), the humidity dependent bias correction in Picarro measured isotope ratios, with standard errors.**

| ORACLES year | Instrument name | Calibration data | $a_D$ (‰) | $b_D$ | $a_{18O}$ (‰) | $b_{18O}$ |
|---|---|---|---|---|---|---|
| 2016 | Mako | 3 laboratory runs over May, 2017 | -0.37 ± 0.056 | 3.03 ± 0.16 | -0.006 ± 0.0015 | 4.96 ±0.21 |
| 2017 | Mako | 4 runs in the field over a 10-day period during ORACLES 2017 | -0.44 ± 0.098 | 2.18 ± 0.29 | -0.013 ± 0.0071 | 3.71 ± 0.53 |
| 2018 | Mako | 1 run in the field during ORACLES 2018 | -0.33 ± 0.085 | 2.86 ± 0.25 | -0.006 ± 0.0056 | 4.58 ± 0.67 |
| 2016 | Gulper | 2 laboratory runs in May, 2017 | 0.035 ±0.008 | 4.46 ± 0.19 | 0.067 ± 0.014 | 1.89 ± 0.20 |

### B.3 Absolute calibration of isotope ratios

Measurements were placed on an absolute scale using several water standards for which the isotope ratio was known (Coplen, 1994). Standard waters (Table B2) were secondary standards based on Florida deionized tap water and "polar" water (a mixture of Antarctic surface snow, mostly from the West Antarctic Ice Sheet). Each was prepared in stainless a steel keg and isotope ratios were measured by the University of Colorado Stable Isotope Laboratory with reference to the International Atomic Energy Agency scale.

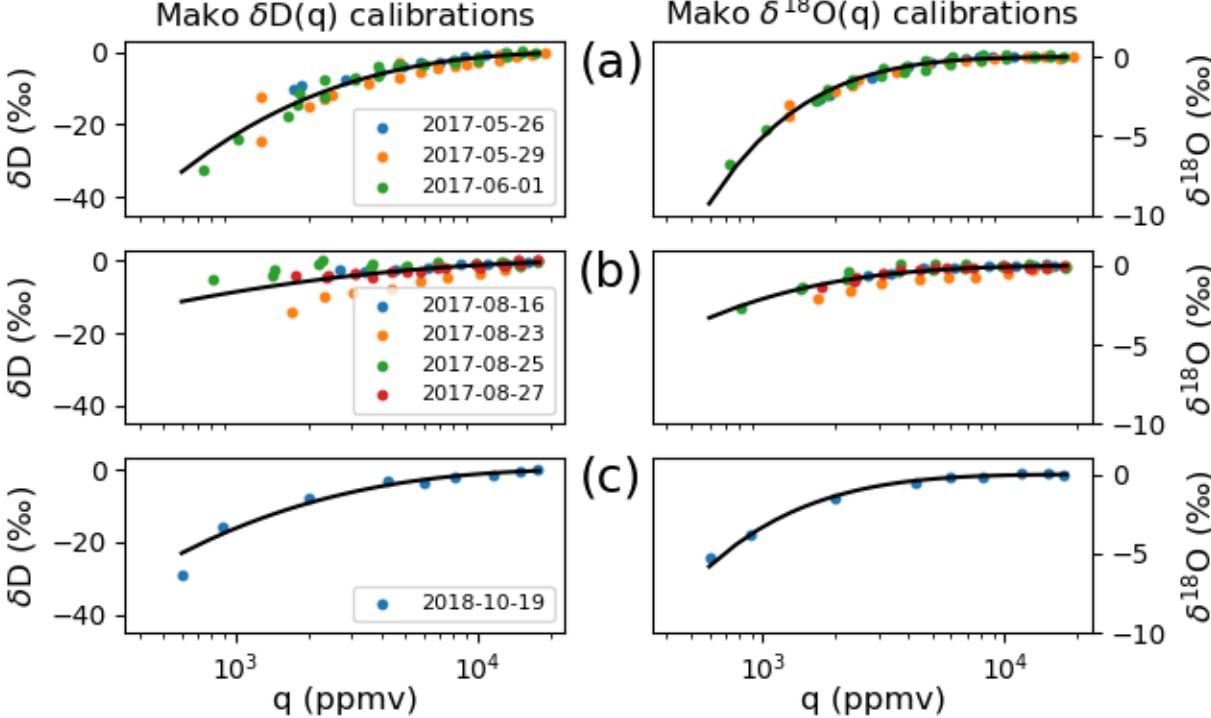

**Figure B1: Mako isotope ratio humidity-dependence corrections for ORACLES (a) 2016, (b) 2017, and (c) 2018. Data points are colored by date that the calibration was performed. Fit curves are generated using nonlinear regression with Eqn. B2.**

Calibrations for each instrument were performed in the laboratory before the first field deployment. Standards were injected by syringe into a Picarro vaporization module and then sampled by the gas analyzers. The relationship between measured and actual δ was taken to be linear from previous field deployments (following, e.g., Noone et al., 2013; Bailey et al., 2016), and a 2-point calibration was performed to obtain the slope and offset. The linear fits are:

$$\delta D_{cal} = 1.056 * \delta D' + 5.96 , \tag{B3a}$$

$$\delta^{18}O_{cal} = 1.052 * \delta^{18}O' + 1.04 + f , \tag{B3b}$$

where the prime superscript refers to data which has had the humidity-dependence correction applied (Eqn. B2). The $f$ in Eqn. B3b is a semi-objective adjustment term (offset), ranging 1-3 ‰, which corrects for observed drift in $\delta^{18}O$ over the 3 years when comparing histograms of P3 data collected below 500 m in altitude. δD does not show this drift. Additionally, deuterium excess ($dxs = \delta D - 8\delta^{18}O$) maxima are anomalously high in 2016 and 2017 when compared to previous studies, in proportion with the $\delta^{18}O$ drift. We have two ideas for the shift. The first is degradation of the optical system resulting from sampling in the highly polluted biomass burning plume (BBA concentrations were higher in 2016 and 2017 then in 2018), with the design of the Picarro optical cavity excluding the possibility for mirrors to be cleaned. The second is aircraft vibrations (which were persistent and at times strong) shaking hardware off of factory specifications. In either case, $\delta^{18}O$ values could be affected more by shifts in instrument hardware than δD since they are ~7x smaller. Previous studies in the Atlantic suggest that the $dxs$ peak is typically observed be between 12-18 ‰ (Benetti et al., 2017), and therefore we chose $f$ to bring $dxs$ into this range, while being faithful to our estimates of absolute calibration.

**Table B2: Isotope standards used for Picarro calibrations.**

| Standard | δD | δ¹⁸O |
|---|---|---|
| Florida tap water | -3.56 ± 0.07 | -0.95 ± 0.06 |
| Antarctica surface snow | -235.3 ± 0.26 | -29.74 ± 0.025 |

## Appendix C: Gulper absolute calibration offset for 2016

The absolute δ calibration for Gulper is assumed to be linear. Therefore, for a measurement in the field by Gulper $\delta_G$:

$$m_G \delta_G + b_G = \delta_{true} ,$$

$$\rightarrow b_G = \delta_{true} - m_G \delta_G , \tag{C1}$$

where $m_G$, $b_G$ are the calibration slope and offset, $\delta_{true}$ is the true value. The slope, $m_G$, is known from calibrations but an estimate of the intercept $b_G$ must be obtained. Due to field constrains on calibration, a direct determination using (C1) that utilized an absolute reference ($\delta_{true}$) was not possible. $\delta_{true}$ is therefore estimated from calibrated Mako measurements by assuming that for a histogram of sub-cloud layer values during similar P3 flight tracks and synoptic conditions, the histogram peak should be roughly the same. By taking $\delta_G$ as the peak for Gulper flights and $\delta_{true}$ as the peak of Mako flights, (C1) is used to estimate $b_G$. The 2016 P3 routine flights (same flight tracks) for Mako were Aug. 31 and Sept. 04. The routine flights for Gulper were Sept. 10, 12, and 25. The histogram peak offsets were 28 ‰ for δD and 6.3 ‰ for $\delta^{18}O$

## Appendix D: Detailed uncertainty estimation

Monte Carlo methods were used to propagate all known uncertainties in the WIA1 parameter fits for Eq. (B2) and (B3), as well as instrument precisions, to produce estimates of total uncertainties for the full range of measured $q$ and isotope ratios. The parameters are uncorrelated and therefore parameter space was sampled using Gaussian distributions centered on parameter expected values and with standard deviations equal to their standard errors. For instrument precision, a Gaussian with standard deviation equal to the instrument precision is sampled. Output was generated over the range of observed $q$ and isotope ratios during ORACLES. Monte Carlo simulations using the ORACLES 2016 parameter fits and errors are used but the results are applicable to all years. Separate simulations of 8,000 iterations each were run for three use cases:

1. Relative comparison of 1Hz data (relative trends in the data, not on absolute scale).
2. Relative comparison of data averaged to 0.1Hz (relative trends in the data, not on absolute scale).
3. Comparison of data averaged to 0.1Hz with other datasets or to theory on an absolute scale.

The uncertainties are a function of both q and the respective δ. For all use cases, the Monte Carlo-derived standard deviations were fit to the following function with $R^2$ of 0.98 or better:

$$\sigma_i = \alpha_0 + \alpha_1 \log(q) + \alpha_2 \log(q)^2 + \alpha_3 \log(q)^3 + \alpha_4 \log(q)^4 + \alpha_5 \delta_i , \tag{D1}$$

where subscript $i$ is for the isotopologue and $q$ is in units of ppmv. The values of the fit parameters $\alpha_0$-$\alpha_5$ are given in Table D1. If the user desires Pic1 uncertainty estimations more detailed than those given in Table 4a, they should compute them using Eqn. D1 and Table D1.

Equation D1 is applicable to WIA1 for the 2017 and 2018 sampling periods and to WIA2 for 2016, but does not account for errors in WIA2 measurements from cross-calibration. It was considered adequate to estimate uncertainties for WIA2 by adding the cross-calibration variance to the WIA1 variance for isotope species $i$:

$$\sigma_{i,WIA2}(q,\delta) = \left[ \sigma^2_{i,WIA1}(q,\delta) + \sigma^2_{i,xcal} \right]^{1/2} , \tag{D2}$$

While the cross-calibration variance $\sigma_{i,xcal}^2$ depends on $q$ and δD, it can be taken as a constant 5 ‰ for δD and 0.4 ‰ for $\delta^{18}O$ (the root-mean-squared-errors from the cross-calibration for $q>4$ g/kg). Since the primary use of WIA2 data is in the MBL (to compare to WIA1 cloud water measurements), those constant values are appropriate for most studies.

**Table D1: Parameter values to use with Eqn. 6 to obtain standard errors in WIA1 isotope ratios, if more detailed error estimations than those given in Table 4a are desired. The three use cases are described in the main text.**

| use case | isotopologue | α0 | α1 | α2 | α3 | α4 | α5 |
|----------|--------------|------|--------|-------|--------|--------|---------|
| 1 | $\delta D$ | 1285 | -522.9 | 80.24 | -5.496 | 0.1417 | -0.024 |
| | $\delta^{18}O$ | 334.4 | -142.6 | 22.91 | -1.64 | 0.0441 | -0.0142 |
| 2 | $\delta D$ | 632 | -267.5 | 42.42 | -2.986 | 0.0788 | -0.0278 |
| | $\delta^{18}O$ | 434.1 | -189.8 | 31.12 | -2.265 | 0.0617 | -0.0169 |
| 3 | $\delta D$ | 568.5 | -242.1 | 38.83 | -2.766 | 0.0738 | -0.0209 |
| | $\delta^{18}O$ | 285 | -125 | 20.63 | -1.511 | 0.0414 | -0.0079 |

*A note on uncertainties for Eq. (3) parameters*

The isotope ratio uncertainty estimations above require values for standard errors on the calibration parameters. While robust estimates of those parameters exist for Eq. (B2), inadequate high quality calibration data was obtained to constrain them for Eq. (B3). Under normal circumstances the calibration slope is robust (e.g., Bailey et al 2016), but between the loose thermistor in 2017 and the drift in $\delta^{18}O$, it is appropriate to provide conservative estimates of $\sigma_{mi}$ and $\sigma_{ki}$ (standard errors in the Eq. (B3) slope and offset for isotope species $i$). For $\sigma_{mi}$, we account for a possible doubling of the deviation of the slopes in Eq. (B3) from unity. For example, the Pic1 slope in $\delta D$ is 1.056 (deviation from unity of 0.056) and therefore we construct a $m_D$ 95 % confidence interval [1, 1+2*0.056] (i.e. $\sigma_{mi}$=0.056/2). For $\sigma_{ki}$, we prescribe different values for the three use cases in the previous section. For the first two, we only care about relative drifts between the years. $\delta D$ did not drift much and so $\sigma_{k,D}$ was assigned 1 ‰. $\delta^{18}O$ on the other hand clearly drifted, and $\sigma_{k,18O}$ was assigned 0.5 ‰ (which is roughly the standard deviation of a $\delta^{18}O$ histogram constructed from data in the sub-cloud well-mixed layer). For the third case, we care about absolute offsets. We assign $\sigma_{k,D}$=4 ‰, $\sigma_{k,18O}$=1 ‰, which give 95 % confidence intervals of +/-8 ‰ and +/-2 ‰ even before including the other parameter uncertainties. Based on variability in marine near-surface measurements (outlined in Benetti et al., 2017 as well as our own), this was considered conservative.

## Data availability

Curtain and vertical profile data for all sampling periods can be accessed at https://doi.org/10.5281/zenodo.5748368 (see Henze et al., 2022). Time series data for the Sept. 2016, Aug. 2017, and Oct. 2018 sampling periods can be accessed at https://doi.org/10.5067/Suborbital/ORACLES/P3/2016_V3, https://doi.org/10.5067/Suborbital/ORACLES/P3/2017_V3, and https://doi.org/10.5067/Suborbital/ORACLES/P3/2018_V3, respectively (see references for ORACLES Science Team, 2020 – 2016 P3 data, 2017 P3 data, and 2018 P3 data). More information on the curtain, vertical profile, and timeseries datasets can be found in sections 5.1, 5.2, and 5.3 respectively.

## Author contributions

DH, DN and DT developed the CVI and WISPER hardware and control software. DN and DH undertook the field deployments with remote support by DT. DH wrote the post-processing and quality control code, selection of the calibration model, and led drafting of the manuscript to which all authors contributed.

## Acknowledgments

This work was supported by a grant from the National Science Foundation Climate and Large-scale Dynamics and Atmospheric Chemistry programs (NSF grant 1564670). ORACLES was a NASA Earth Venture Suborbital-2 investigation managed through the Earth System Science Pathfinder Office. We wish to thank Drs. Jen Redemann, Robert Wood, and Paquita Zuidema for facilitating integration of the isotopic measurements within the ORACLES plan and for coordination with the NSF sponsored activities. Dr. Bryan Rainwater assisted in the development and initial testing of some of the hardware and software components. We acknowledge Dr. Jorgen Jensen of the National Center for Atmospheric Research Aviation Facility for providing guidance and critical scientific and technical

assistance in manufacturing the counter-flow virtual impactor hardware. We similarly are indebted to the staff and
flight crew at the NASA Airborne Science Program based at the Wallops Flight Facility, for assistance with
engineering and logistics support for the measurement program. DN and DH wish to thank experiment
coinvestigators with whom we spent hundreds of hours flying over the South Atlantic with good humor.

**Competing interests**

The authors declare that they have no conflict of interest.

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
