# Peer review of "Aircraft measurements of water vapor heavy isotope ratios in the marine boundary layer and lower troposphere during ORACLES"

_Earth System Science Data, 2021_

## Referee Comment (RC1)

**Review of Henze et al, ESSD**

November 19, 2021

This article presents a dataset of humidity and water vapor isotopic composition during aircraft flights during the ORACLES campaign in the subtropical South-Eastern Atlantic.

The vertical and latitudinal variations in isotopic composition for 3 different years documented by this dataset are unprecedented. I expect that this dataset will be very precious to the isotopic community interested in understanding how boundary layer, cloud, convective and large-scale transport processes impact the water vapor isotopic composition. In particular, vertically-resolved isotopic observations are critically lacking and this dataset fills an important observational gap.

Therefore, I think that this dataset strongly deserves to be published, after a few minor revisions.

**1  Comments on the balance between data, instrument and campaign description, data analysis**

I was asked to specifically comment on whether the balance is right (instrument description, campaign description, versus data description), because ESSD tries to "avoid being a back door for analysis, or instruments". I have several comments and suggestions on this.

- Section 2 describes the campaign and climatology of the sampling region. As a potential data user, I felt this was very useful. However, I think **more information on the altitude of the flight tracks would be very useful**. For example, on fig 1, add colors to indicate altitude along tracks? Or plot the different tracks in a latitude/altitude diagram? Fig 2 gives information about the altitude but I felt this was not enough because we cannot see the connection with the latitude.

- Sections 3, 4 and 5 describe the instrument, calibration details, and uncertainties. I'm not an expert in isotopic measurements, so I'm not able to comment on these sections. I can imagine that measurement experts would find useful information there.

- As a data user, what I am missing at this stage is some section explaining the data format. I went on the links provided in the "Data availability" statement, and it took me some time to understand where to go, and then what archive to unzip, and then which file to look at to find the isotopic data. And once I eventually found the isotopic data, I was still missing the latitude, longitudes and altitudes of the measurements, which are probably described in yet another file? **A paragraph and/or schematic would be useful to guide the users to actually use the data**.

- Section 6 presents some illustration of the data. Maybe this is where the article could be suspected to use ESSD as a "back-door for analysis". As a potential data user, I found this section very useful to be convinced that this dataset provides plausible isotopic signals that can be used for science purpose. The altitude-latitude cross sections and individual profiles allow to have a better idea of the vertical and latitudinal resolutions.
  Yet, **this section could be shortened**, to leave room for a more solid data analysis paper in the future. The angle of this section could be slightly re-oriented to demonstrate the soundness and plausibility of the data.
  Section 6.2.1 is chronologically organized. What I gather from this section is that (1) to first order $\delta D$ is correlated with $q$, especially when looking at altitude and latitude gradients at low levels, and (2) some deviations from this correlation arise because air mass origin also impacts $\delta D$, as supported by its correlation with CO. Maybe this section could be shortened by focusing on these two aspects?

- Finally, I expect that many data users would actually be most interested in the cross sections and profiles illustrated in Figs 7-12. For example, as a modeler who wants to use this data for model evaluation, I would like to have this kind of Level-3 data, without bothering re-doing all the co-location and interpolation work. It is possible to **make this Level-3 data publicly available as part of the dataset**? In this case, section 6 would be even more in line with ESSD editorial policy, because it would describe this Level-3 dataset.

**2   Miscellaneous comments**

- Lines 11-15: The two first sentences could be switched, so that the primary purpose of the article is to describe the dataset rater than the instrument, in better line with ESSD editorial policy.

- Fig 1: what is the unit of the vertical velocity contours? Can you write the numbers for all contours? Also, it would be more intuitive to give values in hPa/d.

- Fig 2 is not referenced in the text. It would be useful in section 2.

- l 406: do you mean fig 7?

- Figs 7, 8, 9: it would be useful to add markers or lines to show the flight tracks, to have an idea what comes from actual observations and what comes from the interpolation.

- l 421: what is misleading? Clarify.

- Fig 7: I don't understand the profiles on the right. Can you clarify the difference between "maximum and typical standard deviation"? Why no dashed line for q?

- Fig 7: why no dot-dashed line as in Fig 8 and 9?

- Fig 8: the blue segment and blue point are arbitrary. I'm not sure they help much.

- Figs 10-12: what do these profiles represent? Are they individual profiles? Or averages over several profiles? What latitudes do they represent? Can the latitude positions be indicated in Figs 7-9?

- l 480: "higher": than what?

- l 481: unclear: do you mean that the standard deviation is three times larger than the instrument precision?

- l 488-489: I don't understand this sentence. "evident in the figure" should also be replaced by a more specific and objective statement.

---

## Author Response (AR1)

**Author's response to Referee #1's review of Henze et al, ESSD**

1/31/2022

**1 Comments on the balance between data, instrument and campaign description, data analysis**

I was asked to specifically comment on whether the balance is right (instrument description, campaign descrip- tion, versus data description), because ESSD tries to "avoid being a back door for analysis, or instruments". I have several comments and suggestions on this.

**Thank you for your constructive feedback. We have made some adjustments ensure that emphasis is on the dataset, recognizing the scientific analysis is presented elsewhere (a paper soon to be submitted to ACP). We respond to each of your bullets below.**

1.1) Section 2 describes the campaign and climatology of the sampling region. As a potential data user, I felt this was very useful. However, I think **more information on the altitude of the flight tracks would be very useful**. For example, on fig 1, add colors to indicate altitude along tracks? Or plot the different tracks in a latitude/altitude diagram? Fig 2 gives information about the altitude but I felt this was not enough because we cannot see the connection with the latitude.

**We are glad you found Section 2 useful. Aircraft latitude/altitude tracks have been added to Fig. 2 as suggested. The reader can now quickly and easily identify the sampling coverage for each observation period.**

1.2) Sections 3, 4 and 5 describe the instrument, calibration details, and uncertainties. I'm not an expert in isotopic measurements, so I'm not able to comment on these sections. I can imagine that measurement experts would find useful information there.

**We placed portions of sections 3, 4, and 5 in appendices. We feel it is important to still include the information, if only in appendices, since to our knowledge the isotope community is still in the process of standardizing aircraft measurement and calibration procedures so that datasets are comparable across different observational campaigns.**

1.3) As a data user, what I am missing at this stage is some section explaining the data format. I went on the links provided in the "Data availability" statement, and it took me some time to understand where to go, and then what archive to unzip, and then which file to look at to find the isotopic data. And once I eventually found the isotopic data, I was still missing the latitude, longitudes and altitudes of the measurements, which are probably described in yet another file? **A paragraph and/or schematic would be useful to guide the users to actually use the data**.

**Sections 5 and 6 have been overhauled and merged into a single section 5 which now describes the three WISPER data formats available to the user (and includes the illustrative examples originally in section 6). We have taken your suggestion below and made available Level 3 dataset for curtains and vertical profiles, which sections 5.1 and 5.2 describe. Section 5.3 describes the time series data (originally the only dataset offered) and includes a schematic to navigate to the files from the directories point to by the DOIs. The files we direct the users to in this schematic are files which already include latitude, longitude, and altitude.**

1.4) Section 6 presents some illustration of the data. Maybe this is where the article could be suspected to use ESSD as a "back-door for analysis". As a potential data user, I found this section very useful to be convinced that this dataset provides plausible isotopic signals that can be used for science purpose. The altitude-latitude cross sections and individual profiles allow to have a better idea of the vertical and latitudinal resolutions.

Yet, **this section could be shortened**, to leave room for a more solid data analysis paper in the future. The angle of this section could be slightly re-oriented to demonstrate the soundness and plausibility of the data.

Section 6.2.1 is chronologically organized. What I gather from this section is that (1) to first order $\delta D$ is correlated with $q$, especially when looking at altitude and latitude gradients at low levels, and (2) some deviations from this correlation arise because air mass origin also impacts $\delta D$, as supported by its correlation with CO. Maybe this section could be shortened by focusing on these two aspects?

**Thank you for the suggestions. We share the view that ensuring the user has confidence via a plausibility check is important. The discussion in section 6 (now in section 5) has been shortened to emphasize your points (1) and (2). Additionally, qualifying statements are included, emphasizing that the discussion in this section is meant to assess the soundness and plausibility of the data (as you pointed out), but also hint at interesting aspects of the data that could be explored in future studies.**

1.5) Finally, I expect that many data users would actually be most interested in the cross sections and profiles illustrated in Figs 7-12. For example, as a modeler who wants to use this data for model evaluation, I would like to have this kind of Level-3 data, without bothering re-doing all the co-location and interpolation work. It is possible to **make this Level-3 data publicly available as part of the dataset**? In this case, section 6 would be even more in line with ESSD editorial policy, because it would describe this Level-3 dataset.

**We have made Level-3 curtain data and vertical profiles available as .nc files hosted on Zenodo and created a DOI with the link presented in the revised manuscript. Section 5 now includes subsections describing these datasets.**

**2 Miscellaneous comments**

2.1) Lines 11-15: The two first sentences could be switched, so that the primary purpose of the article is to describe the dataset rather than the instrument, in better line with ESSD editorial policy.

**The first two sentenced have been switched.**

2.2) Fig 1: what is the unit of the vertical velocity contours? Can you write the numbers for all contours? Also, it would be more intuitive to give values in hPa/d.

**All red contours (aerosol optical depth) now have labels. The shaded vertical velocity are only shown in order to demonstrate that sampling took place in a region of large-scale subsidence, and therefore only the sign is important here (this point has been added to figure caption). The only vertical velocity contour that is present in the figure is the boundary between shaded (upward velocity) and non-shaded (subsidence) regions.**

2.3) Fig 2 is not referenced in the text. It would be useful in section 2.

**Fig. 2 is referenced in section 2 of the original text (line 95). After revisions it is now referenced twice in section 2.**

2.4) l 406: do you mean fig 7?

**Yes thank you, we meant to reference Fig. 7, but this sentence has been deleted in the revision.**

2.5) Figs 7, 8, 9: it would be useful to add markers or lines to show the flight tracks, to have an idea what comes from actual observations and what comes from the interpolation.

**Fig. 2 has been amended to include aircraft latitude/altitude tracks for each observation period, the reader should now have an idea of what parts of Figs. 7-9 are from observations vs. interpolation. The actual number counts are provided in the curtain datafile provided. The lat/alt coverage for this campaign is substantial, and consequently make Figs. 7-9 too busy to see the data contours, and we have opted not added them to those figures.**

2.6) l 421: what is misleading? Clarify.

**This sentence is deleted in the revision.**

2.7) Fig 7: I don't understand the profiles on the right. Can you clarify the difference between "maximum and typical standard deviation"? Why no dashed line for q?

**The objective is to provide information on typically observed standard deviations (i.e., expected fluctuations) within the cross section domain. To better capture this, we have opted instead to plot standard deviation contours overlain on the curtains. This has the benefits of (1) being more intuitive, and (2) gives more detail on the latitudinal structure of the variabilities.**

2.8) Fig 7: why no dot-dashed line as in Fig 8 and 9?

**A dot-dashed line has been added to Fig. 7. We also used this as an opportunity to update the dot-dashed lines in Fig. 8 and 9 to reflect a refined method to identify MBL tops.**

2.9) Fig 8: the blue segment and blue point are arbitrary. I'm not sure they help much.

**The blue segment and point have been removed.**

2.10) Figs 10-12: what do these profiles represent? Are they individual profiles? Or averages over several profiles? What latitudes do they represent? Can the latitude positions be indicated in Figs 7-9?

**The text now clarifies that Figs 10-12 are examples of individual profiles. We have also made this more explicit in the figure captions. The captions now also have the latitudes at which the profiles were taken, and markers are placed on Figs. 7-9 (now Figs. 6-8) corresponding to these latitudes.**

2.11) l 480: "higher": than what?

**The discussion here has been changed and this sentence is no longer included.**

2.12) l 481: unclear: do you mean that the standard deviation is three times larger than the instrument precision?

**The discussion here has been changed and this sentence is no longer included.**

2.13) l 488-489: I don't understand this sentence. "evident in the figure" should also be replaced by a more specific and objective statement.

**The discussion here has been changed and this sentence is no longer included.**

**Author response to Referee #2's review of**

**"Aircraft measurements of water vapor heavy isotope ratios in the marine boundary layer and lower troposphere during ORACLES"**

**by Henze, Noone and Toohey,**

**ESSD paper number essd-2021-238.**

========

Summary:

The authors present measurements of water and its isotopic composition in the lower troposphere obtained during the ORACLES field campaign. In the first half of the paper, a brief description of the campaign and the place of isotopes within ORACLES is followed by extended descriptions of (1) the measurement system used to feed samples of water to commercially-produced instruments that measure isotopes and (2) the calibration of a reference instrument against standards of known composition and of a second instrument against the first. Last, some of the data itself is presented.

===========

Assessment:  Major revisions

As the authors note, this dataset is new and exciting, as it shows the vertical structure of isotopic composition in the lower troposphere over the southeast Atlantic and its variation with latitude.  This dataset should be published, but I have several suggestions below that I believe could help the reader better understand the dataset and its properties.  The main suggestion is to either add cloud water isotopic data to section 6 or to remove the extended discussion of the CVI inlet and its plumbing since it is only relevant to the cloud water measurements.  In addition, I have several suggestions that I would ask the authors to consider in their revisions.  At the bottom, optional stylistic suggestions are also made.

===============

1. Major comment:

1.1) The measurements in the dataset are of total water (vapor + condensate) and cloud water, yet no observations of cloud water are shown in section 6.  As far as I could tell, section 6 also makes no mention of whether the measurements presented in figures 7-12 are of water vapor or total water.  If someone wanted to use the cloud water data included in the dataset, this paper does not seem to present the data itself, so I would encourage the authors to choose one of the following two options:

 - shorten the paper substantially by removing the discussion of the CVI (and possibly also the cross-calibration of the two analyzers), since they are not relevant unless the cloud liquid measurements are presented here, or

 - add at least one figure showing coincident measurements of cloud liquid and total water mass and isotopic composition, as well as how far the cloud water isotopic composition is from isotopic

equilibrium with the surrounding vapor. Note that this can be done using the total water isotopic composition and the ratio of cloud liquid mass to vapor mass.

**These are great points and we have decided to add a figure which shows**

    **(1) Two example individual profiles showing LWC, and vapor vs. cloud dD and dxs with theoretical equilibriums.**

    **(2) Probability distributions of observed dD and d18O fractionation factors for all cloud measurements, each overlain with probability distributions of equilibrium fractionation computed from the in-situ temperature measurements.**

    **It has also been made clear that the curtains and individual profiles in Figs 7-12 (now Figs 11) are for total water quantities.**

1.2) It would be helpful if there was a standard reference about the calibration of water vapor isotope analyzers, so that the authors could refer to that and only highlight the details where their approach differs from that reference. I suppose each scientist or group has their own techniques, so that they prefer their own approach. This isn't a complaint about this paper but a general lament.

**We adjusted the text to better align the manuscript with your comment, at least for water isotope analyzer 1 (WIA1). In section 4.3, which originally had the lengthy description, tables, and figures, for WIA1 calibration, we have instead added a short paragraph giving the overview of the calibration procedure and sighting references for each step. While there is some degree of maturity in the community on how calibration should be done, at ground level, there is no standard method. Similarly, airborne measurements of isotope ratios are sufficiently new that there is not standard approach, in part due to the different possible details of flight operations. Therefore a detailed description is still included in an appendix. For WIA2 on the other hand, the crosscalibration procedure was novel and there is no literature to reference as far as we know.**

========================================================================

2. Things that should be considered and responded to before publication:

2.1) section 3: I understand the structure of this section, dividing the presentation into 1) inlets, 2) flow configuration, 3) transfer lines, 4) heating of transfer lines/inlets, and 5) analyzer details. However, the section is really dominated by the extended discussion of the CVI inlet, so I would suggest presenting the full description of the SDI inlet (and its transfer line and heating system) before doing the same for the CVI inlet. This would have the benefit of emphasizing that the SDI inlet brings both droplets and vapor together and allows the droplets to evaporate before the resulting vapor (which reflects total water) is ingested into the isotope analyzer. When the treatment of the CVI piping/heating is similar to that of the SDI, this could be noted, making the CVI description shorter.

**Section 3 has been restructured and now the SDI and CVI share roughly equal real estate. The first paragraph is a brief intro, then paragraph two presents the SDI, and paragraph 3 presents the CVI. This shortened the section, allowing room for a brief description of the CVI workings as suggested in your comment 2.3).**

2.2) l111: Please do not use an abbreviation that comes from a brand name of a commercial product (here, Pic is short for Picarro, I assume).  I would suggest something more generic: WVIA1 and WVIA2.

**The abbreviations have been changed to WIA1 and WIA2 in this manuscript. However, for the .ict datafile headers and .nc file variable explanations they remain Pic1 and Pic2 and this is emphasized in the manuscript both in sections 3 and 5. While we agree with the sentiment of the comment, we do not change the labels for two pragmatic reasons: (1) the DOI containing our dataset also contains 10+ other datasets, and whenever we change an aspect of our dataset the data managers in charge of the ORACLES dataset must remake mission merged files in a manner in which versions can be tracked back to field collected data, which has original labels; and (2) For the datafiles, it seems reasonable to allude to the brand of instrument used since it is relevant technical information and reflects the specific hardware choices used for the campaign.**

2.3) l101-102, 129-131 and 176: Please string a few sentences together here to tell the reader how the CVI system generates a measurement of cloud liquid mass and isotopic composition.  Include in this an explanation of the CVI enhancement factor.  Only a few sentences are needed, but please carry the reader from inflow to the separation of cloud droplets and the surrounding vapor, to the evaporation of those droplets in the dry counterflow air, and from there to the isotope analyzer.  Make the reader understand why this procedure leads to a robust measurement of cloud liquid isotopic composition.

**In the original manuscript there is a one sentence explanation of the CVI, while the revision has been expanded to:**

**"The CVI, adapted from the NSF Gulfstream-V inlet (G-V CVI), separates the condensed water particles (liquid + ice) from the ambient vapor. The CVI works by pushing dry air counterflow, opposite the direction of the inlet sampling flow, through holes in the CVI tip which prevents the ambient air from entering the inlet. The counterflow strength is tuned so that condensed water particles are able to pass via their inertia. The CVI tip and transfer lines are heated so that water particles evaporate completely before sampling by the isotope analyzers. The CVI is operated at sub-isokinetic flow, leading to an enhancement of the water particle volume concentration within the transfer lines. Since the isotope analyzers have increased precision at higher water concentrations, a typical enhancement factor of 30 allows for robust measurements in clouds with liquid water content as low as 0.04 $kg/m^3$ and science-usable measurements for lwc down to 0.01 $g/m^3$."**

2.4) Also emphasize that the WVIA1 is almost always pulling from the same air stream that WVIA2 is measuring and that only inside cloud is WVIA1 switched to the CVI line.  This was my understanding from later in the paper.  Is this true?

**Yes, WIA1 and WIA2 are both on the SDI for the majority of each flight. A sentence has been added to section 3, paragraph 2 to clarify this.**

2.5) l131-139: The comparisons with the NCAR GV CVI system sound strange here unless there is an existing reference on the GV CVI system that you can reference here.  I would suggest starting by saying that the CVI system design was based on that of the NCAR GV and then just describing how it works before mentioning the appendix.  The comparisons make sense in the appendix but not here.

**The comparison to the GV CVI has been removed from the main text as suggested.**

2.6) l140-147: Introduce the three water vapor isotope analyzers first, then talk about their positions in WISPER (i.e., WVIA1 and WVIA2). Also, I would suggest always going in chronological order from 2016 to 2017-18, unless there's a really good reason not to.

**We reorganized this paragraph to flow in chronological order. The three water isotope analyzers are introduced in the order they appear (chronologically, then by WIA1 and WIA2 position), rather than introduced all at the beginning. We feel this has a good flow and is easy to follow.**

2.7) l243-252: The fudge factor, f, in the calibration of delta18O does not inspire great confidence in the data, even if it is necessary. The accumulation of aerosol on the mirrors within the cavity is a plausible explanation, but could the authors suggest why this affects only delta18O and not deltaD and water vapor? As the Benetti data came from the surface layer, it may not exactly apply to the aircraft data at higher altitudes, though I do imagine that vertical gradients in deuterium excess would not be strong.

**Agreed. It arises, fundamentally, as a pragmatic solution to the calibration problem that result from shortcomings of field calibration that were performed. One must recall that that d18O values are typically ~x8 smaller I nmagnitude (closer to zero permil) than dD linked to much lower natural abundance) and therefore may be affected fractional error in either measurement or data handling during calibration. More over the 18O spectroscopy is known to be more sensitive, again in part to natura abundance, but also linked to the specific line strengths within he scanned absorption range. We have added this idea into the manuscript. We have also included the possibility that aircraft vibrations shifted calibration.**

**Since the Benetti data came from the surface, we only compared our values at altitudes below 500 m. We are only looking for a modification to the offset and not the slope, this point-comparison is sufficient for a modification to the calibration offset, which is the target of the f-term.**

2.8) l268, eqn 4, fig 5-6: This calibration polynomial fit (fifth order in each of log(q), delta and log(q)*delta) seems very complicated, and its presentation in figure 5-6 makes it hard for the reader to understand the motivation for the form suggested in equation 4 and the quality of the calibration. My first suggestion would be to calibrate WVIA2 against WVIA1 in the same way that WVIA1 was calibrated against the standard. This would require little text to explain, and the coefficients of the fits could be presented compactly, perhaps by adding them to the tables with the coefficients for WVIA1.

As the authors have already produced a data set based on equation 4, I expect that they are reluctant to re-do the cross-calibration of WVIA1 and WVIA2. I have a couple of suggestions to make equation 4 and figures 5-6 more clear for the reader. First, equation 4 should have delta2_calib on the left hand side and not delta1. If the authors want to keep delta1 on the left hand side, an error term E_calib should be added to the right hand side because the polynomial fit will not in general match the value of delta1. Second, the quality of the calibration will be shown better in figures 5-6 by plotting this error (E_calib = delta1 - delta2_calib), rather than the calibration curves. I would suggest putting log(q1) and delta1 on the x- and y-axes since they are the reference and binning the data into delta1-log(q1) bins with about 10-15 bins in each direction. The RMS error of delta1-delt2_calib in each bin could be shown with the

colorscale.  Bins with no data could be left blank (by filling them with NaNs in MATLAB or whatever is the equivalent in python).

**We concluded that the WIA2 cross-calibration needs to be a function of its own measured water concentration (q) and isotope ratio (delt) due to the fact that WIA1 is periodically switching to the CVI. For those time intervals, there would be no input values for the cross-calibration function if it required q and delt from WIA1, and these are the intervals when WIA2 data is desired the most.**

**We therefore keep the cross-calibration formula in terms of WIA2 values, but add the error term as recommended. The cross-calibration figures have also been changed as suggested: log(q1) and delta1 are on the x- and y-axes, and the plot is a colormap of RMS error.**

**During this revision, we found that our estimates for the 2017 dD cross-calibration RMS were higher than previously thought, so table 4b has been refined accordingly.**

2.9) l296-311: This is more of a list than a paragraph.  Perhaps it could be shifted to a bulleted list in an appendix.

**As a style choice, we're kept this as a narrative form. We feel the information in this paragraph is of primary importance to users of the time-series data and therefore belongs in the main text.**

2.10) l334-350, eqn 5 and table 5: A reader will want to look at table 5 and be able to read off the uncertainty in at least one value of q/delta and see how the uncertainty varies around that value.  The formulation of equation 5 makes this difficult at best.  I would suggest finding the most common combination of q and delta in the dataset, calling them q_ref and delta_ref and then replacing q and delta in equation 5 with q/q_ref and delta-delta_ref.  Then, alpha0 will give the value of the uncertainty for q/delta = q_ref/delta_ref and alpha1/alpha5 will show how they vary linearly with log(q) and delta around that point.

**For most users of the data (e.g. satellite validation, studies involving mean curtains or profiles), the uncertainties quoted in tables 4a and 4b are sufficient, especially sincenow that we have added two data products for users: average latitude-altitude curtains, and individual vertical profiles averaged onto 50 m levels.**

**Only a small fraction of users are likely to implement eqn 5. (e.g. those looking at detailed time series analysis or detailed vapor liquid comparisons) and at the point where they are delving into eqn 5, we feel that using it as-is vs. a q/q_ref version is not a major deterrent. To reflect this, we have moved eqn 5 and table 5 to an appendix D.**

2.11) section 6: Three thoughts:

 - If the authors are encouraging users to average the 1 Hz data to 10s intervals, perhaps they should consider providing a 0.1 Hz dataset.  If nothing else, it would reduce the dataset size by a factor of ten for users who were not interested in the 1Hz data.

**We appreciate this very reasonable point. The timeseries data are a small part of the overall ORACLES dataset with 10+ other instruments, and the standard was that all instruments**

**interpolated to a common 1 Hz time step. Therefore, it makes sense to keep them this way, particularly for the merged timeseries files.**

**However, for our revised manuscript we have added two new data products hosted independently of the ORACLES timeseries collection. We have added: (1) mean latitude-altitude curtains for each sampling period, and (2) Individual vertical profiles averaged onto 50 m vertical levels. We think that the majority of users will use these two data products over the timeseries files, so in a roundabout way this is a solution to your comment.**

- Does q in the figures, text and tables correspond to total water or water vapor? This should be specified somewhere.

**It is now specified in the text that these figures are for total water.**

- If the discussion of the CVI is maintained in the revised manuscript, some cloud water data should be presented here. A couple of suggestions are made above.

**A figure for CVI data has been added as detailed above..**

l388-394: Some of the differences between measurements in 2016 and those in 2017-18 could be related to the different regions that were observed then.

**We agree with your point and our idea was to highlight the possible differences in hydrology of the regions leading to the different signals. Regardless, this paragraph has been removed from the revision, as sections 5 and 6 have been substantially overhauled.**

2.12) sec 6.2.1: Since the composite curtains are based on water-mass-weighted values, the isotopic composition will be skewed towards its value in samples with high q. This probably matters most in regions where the variability of q is largest. In particular, variations in the height of the trade inversion probably leads to the largest amount of variability in q. I would speculate that this could contribute to the unexpectedly low correlation between q and dD in the composites for 2017 and 2018.

**Yes, this does seem to be the case (and seen in more detailed analysis of the data). The trade inversion height variability leads to some of the largest q variability. It need not lead to the low correlation, however, , since the standard deviation in q at trade inversion top (3 g/kg) is equal to or higher than those for 2017 and 2018. There is also a key point here on the use of mass weighted statistics vs simpler expectations (non-weighted). Both have utility and merit depending on what types of questions are asked. (Mass weighting is not always appropriate).**

2.13) l411: Could a brief explanation (only a half or full sentence) be added to make clear why "it's clear" that the moisture and dD come from the the continental PBL? This might not be obvious to the uninitiated reader.

**A brief explanation has been added, connecting the CO to evidence of biomass burning from the African continent (and references have been added).**

2.14) Figures 7-9: Is there some way to specify what inside/outside the plume? This is less obvious in figures 8 and 9. My suggestion would be to close the contour with a thick line on the right edge of the panels in figure 7, with similar thick lines bounding the edges of the plume in figures 8-9.

**Two sentences have been added to the main text and to the caption of Fig. 7 (now Fig. 6) specifying that the CO contour bounds the 2 – 5 km region where biomass burning plumes were typically sampled.**

2.15) Figures 10-12: A couple of suggestions:

**Figures 10-12 are now figures 9 -11.**

 - It would be useful to add a thin line (perhaps dashed or dotted) to show the y-axis for q. Otherwise, it is difficult to gauge how small q becomes at higher levels.

**Lines have been added.**

 - Give the latitude, longitude and time of day (or range of those values) when the profile was sampled. The interested reader might wish to look at satellite images or reanalysis to learn about the meteorological situation when they were sampled.

**Mean latitude, longitude, and time have been added to the figure captions.**

2.16) l484-486: Is it possible that the ingestion of cloud liquid has a lasting effect after passing through a cloud, or that the differing timescales of dD and d18O might lead to such excursions of deuterium excess (Aemisegger et al, 2012, ACP, https://doi.org/10.5194/amt-5-1491-2012, sec. 7)? On this point, I found myself wondering whether there were any systematic differences between isotopic measurements during upward and downward profiles.

**While we did observe some systematic differences between upward and downward profiles, they were on smaller length scales (100 – 200 m) and more drastic changes in water concentration (instances where q dropped to $\prec$ 1 g/kg) than in Fig. 11b (now 10b). A note on this has been added to the text.**

2.17) In addition, the large deltaD excursion just above the PBL in figure 12b surprised me because the air was only slightly drier than the subcloud layer, so I didn't understand how it could be so depleted. Perhaps, I am missing the key physical process going on there.

**In the original manuscript we attributed this simply to "potential precipitation", while more detailed quantitative analysis of the data suggests that a Rayligh process alone is insufficient. Rain re-evaporation in one mechanism that could further deplete dD (see e.g. Noone 2011). In fact an analysis done in forthcoming work points to rain evaporation. This point has been added to the manuscript.**

=========================================================

3. Wording/stylistic suggestions that are not essential:

3.1) l55: Since ORACLES is defined in the abstract, the abbreviation could be used here without writing out the full name.

**The full ORACLES name is kept here. Just a stylistic choice.**

3.2) l80: "... in the strong inversion _atop_ MBLs _in this_ region."  Also remove "then" before "transition".

**Sentenced revised.**

3.3) l91-92: "... every 2-3 days, lasted 7-9 hours and occurred during daytime hours (from 7am to 5:30pm local time)."

**Sentenced revised.**

3.4) l175/table 1: If the transit flights are not included in the dataset, maybe they don't need to be in table 1.

**Rows removed**

3.5) l176: Explain CVI enhancement factor if it wasn't already done above.

**An explanation of the enhancement factor is included further up as per one of your suggestions in the previous section.**

3.6) l177: Define deuterium excess here if not done elsewhere.

**Since deuterium excess is not explicitly included as one of the variables in the dataset, we save its definition till section 6.**

3.7) l185-190: The chunk of long instrument names upsets the flow of this paragraph.  I would suggest moving the explanation of the coincident spikes ahead of the instrument names, as in "Time synchronization to the cloud probes was achieved indirectly by aligning concident spikes in biomass burning aerosol, carbon monoxide and water vapor specific humidity, as measured by the ... (PCASP), COMA instrument (...) and the two water vapor isotope analyzers, respectively."

**We agree that the flow of this paragraph could be improved. The specific wording suggested above was tried and didn't seem to improve the paragraph, so we used an alternate re-wording which we feel has good flow.**

3.8) l201: There must be a good reference for the way the Picarro measures water vapor.  Please add a citation to it.

**Citation added.**

3.9) figure 3: Maybe add an arrow and text to indicate that the curved black line is the P3 fuselage.

**Arrow and text added to figure 3.**

3.10) l217-221: Move the "Table 2 ..." sentence ahead of "The calibration parameters do not change ...".  I think it's important for the reference to the table and figures to be close to the equation.

**Sentence moved.**

3.11) l388: "... over the African _continent_ contains ..."

**Changed.**

3.12) l406: Figure 7 is being discussed here, not the present figure 8.  Also, figure 10 should be moved forward, so that the figures are ordered corresponding to when they appear in the text.  This keeps the reader from having to flip back and forth a lot.

**Figure reference changed to Fig. 7. Also, Fig. 10 has been moved forward and subsequent figures renumbered.**

3.13) l409-419: The MBL is part of the lower troposphere.  When referring to the region above the MBL, I would suggest using "lower free troposphere" or LFT.

**"lower free troposphere" is now used when first introduced, and "LFT" is used in all further mentions thereafter.**

3.14) l474: "colder, drier air" rather than "cold, dry"

**Changed.**

3.15) l516-517: "... several instances of layers just above the MBL whose dD values are more depleted that the air above and below."

**Changed.**

3.16) l559: "of the feedback loop" is unnecessary.

**This phrase has been kept in since it doesn't seem to be problematic.**

**References:**
**Noone, D. "Pairing Measurements of the Water Vapor Isotope Ratio with Humidity to Deduce Atmospheric Moistening and Dehydration in the Tropical Midtroposphere." *Journal of Climate*, vol. 25, no. 13, 2012, pp. 4476–4494. *JSTOR*, www.jstor.org/stable/26192015.**